# OPTIMAL STOPPING VS BEST-OF-$N$ FOR INFERENCE TIME OPTIMIZATION

## ABSTRACT

Large language model (LLM) generation often requires balancing output quality against inference cost, especially when using multiple generations. We introduce a new framework for inference-time optimization based on the classical Pandora's Box problem. Viewing each generation as opening a costly "box" with random reward, we develop algorithms that decide when to stop generating without knowing the underlying reward distribution. Our first contribution is a UCB-style Pandora's Box algorithm, which achieves performance that is provably close Weitzman's algorithm, the optimal strategy when the distribution is known. We further adapt this method to practical LLM settings by addressing reward scaling across prompts via a Bradley–Terry inspired transformation. This leads to an adaptive inference-time optimization method that normalizes rewards and learns stopping thresholds on the fly. Experiments on the AlpacaFarm and HH-RLHF datasets, using multiple LLM–reward model pairs, show that our adaptive strategy can obtain the same performance as non-adaptive Best-of-$N$ sampling while requiring 15-35% fewer generations on average. Our results establish a principled bridge between optimal stopping theory and inference-time scaling, providing both theoretical performance bounds and practical efficiency gains for LLM deployment.

## 1 INTRODUCTION

Large language models (LLMs) are increasingly deployed in applications where both quality and efficiency are critical (Achiam et al., 2023; Hoffmann et al., 2022). A widely used approach to improve generation quality is Best-of-$N$ sampling: generate $N$ candidate responses, score them with a reward model, and select the best (Nakano et al., 2021; Touvron et al., 2023). While effective, this approach wastes compute: the number of generations is fixed in advance, even if an acceptable output is found early or if a prompt is inherently easy (Wang et al., 2025; Sun et al., 2024; Manvi et al., 2024). As models scale and inference costs rise, the need for adaptive inference-time strategies that dynamically balance quality and compute has become urgent (Snell et al., 2025; Jin et al., 2025).

Several recent methods have sought to improve inference-time efficiency. Reranking strategies (e.g., Best-of-$N$, rejection sampling, majority-vote) improve quality but rely on over-generation, making them computationally expensive (Wang et al., 2025; Manvi et al., 2024; Jain et al., 2023; Wang et al., 2022). Speculative decoding accelerates sampling by offloading work to a smaller draft model, but it does not address how many generations to produce (Leviathan et al., 2023; Chen et al., 2023). Early stopping heuristics exist in practice, yet they lack theoretical guarantees and often underperform on difficult prompts (Agrawal et al., 2024; He et al., 2025; Wei et al., 2025). Overall, there lacks a principled framework for deciding when to stop generating while maintaining near-optimal reward.

In this paper, we introduce a new perspective by connecting LLM inference with the classical Pandora's Box problem from optimal stopping theory (Weitzman, 1978). In our view, each generation corresponds to opening a costly "box" that yields a random reward. The task is to decide whether to stop and accept the best reward so far or continue generating, *without knowing the underlying reward distribution*. This abstraction provides a rigorous foundation for inference-time optimization, subsuming heuristics like Best-of-$N$ as special cases.

**Our Contributions.** We develop both theoretical and practical foundations for adaptive LLM inference on the following three fronts:

1. **A UCB-style Pandora's Box algorithm.** We propose the first stopping strategy that adapts to unknown reward distributions. By maintaining anytime-valid confidence bounds on the optimal stopping threshold, our algorithm guarantees vanishing regret relative to Weitzman's optimal policy, which assumes full distributional knowledge. [1]

2. **A practical adaptive meta-generation framework.** To handle cross-prompt reward scaling issues, we introduce a Bradley–Terry inspired transformation that normalizes rewards. This yields a general-purpose meta-generation procedure that dynamically learns stopping thresholds.

3. **Empirical validation.** On the AlpacaFarm and HH-RLHF dataset, across multiple LLM–reward model pairs, our adaptive strategy achieves the same reward as non-adaptive Best-of-$N$ sampling while requiring 15-35% fewer generations on average. This establishes that principled stopping rules yield concrete efficiency gains in realistic settings.

## 1.1 RELATED WORKS

**Inference-Time Optimization.** A growing body of work studies how to allocate inference-time compute more effectively. Beyond Best-of-$N$, (Nakano et al., 2021; Touvron et al., 2023; Wang et al., 2025; Sun et al., 2024; Manvi et al., 2024), recent methods frame test-time scaling as adaptive self-calibration (Huang et al., 2025; Qu et al., 2025). Chain-of-thought prompting and self-consistency (Wei et al., 2022; Wang et al., 2022) also improve reliability by generating multiple reasoning paths, though at the cost of substantial extra compute. Lastly, Snell et al. (2025) show that prompt-adaptive compute allocation can outperform model scaling, while Jin et al. (2025) analyze energy–accuracy tradeoffs. See the survey by Welleck et al. (2024) for a comprehensive review.

**Early Stopping, Confidence, and Adaptive Decoding.** Several works address the question of *when to stop* allocating further computation. Classic approaches include adaptive computation time (Graves, 2016) and confident adaptive decoding (Schuster et al., 2022). Agrawal et al. (2024) propose an entropy-based stopping rule during speculative decoding, adaptively halting draft expansion when confidence is sufficient. He et al. (2025) introduce an uncertainty-guided mechanism for code generation, pausing and reranking when uncertainty is high. Similarly, Wei et al. (2025) leverage adaptive layerwise exits to accelerate decoding without sacrificing accuracy. These approaches share our motivation of minimizing unnecessary compute while preserving output quality.

**Pandora's Box and Optimal Stopping.** Our approach draws inspiration from the *Pandora's Box problem*, a classic framework in optimal stopping (Weitzman, 1978). Recent advances have expanded this framework into online and learning-theoretic domains Esfandiari et al. (2019); Gergatsouli & Tzamos (2022); Atsidakou et al. (2024), establishing important connections with prophet inequalities and multi-armed bandit formulations (Gatmiry et al., 2024; Xie et al., 2024). While this problem has generated substantial theoretical insights across various domains, its application to LLM inference remains unexplored. Our work brings this perspective to test-time optimization by framing candidate generation as opening costly boxes.

We review other related works in Appendix B.

## 2 PRELIMINARIES

### 2.1 NOTATION

Let $\mathcal{X}$ denote the space of prompts and $\mathcal{Y}$ be the space of responses. A Large Language Model (LLM) $\pi : \mathcal{X} \to \Delta\mathcal{Y}$ maps a prompt to a distribution over responses, where we let $\Delta\mathcal{Y}$ denote the set of all distributions on $\mathcal{Y}$. A reward model is a function $r : \mathcal{X} \times \mathcal{Y} \to \mathbb{R}$ that maps a prompt and a response to a real-value. Given a prompt $x \in \mathcal{X}$, LLM $\pi$, and reward model $r$, we use $D_{r,\pi(x)}$ to denote the distribution over rewards induced by passing $x$ to $\pi$, sampling $y \sim \pi(x)$, and then computing $r(x, y)$. We will often just use $D$ as shorthand and use $F$ to denote its CDF and $f$ to denote its PDF, omitting dependence on $\pi$, $x$ and $r$ when those are clear from context.

---

[1] We note, however, that Weitzman's algorithm applies to multiple box types. Our results are concerned with the i.i.d. special case, though we expect effective the natural generalization of our approach to multiple boxes will apply to generation from LLM ensembles

## 2.2 TEST-TIME STEERING AND BEST-OF-$N$ SAMPLING

After the pre-training stage, the focus often shifts from learning general language ability to steering a model's outputs toward more desirable responses. One way to formalize this is through a reward model $r : \mathcal{X} \times \mathcal{Y} \to \mathbb{R}$, which assigns higher scores to responses $y$ for a given prompt $x$ when they exhibit preferred properties. The objective is then to generate outputs that achieve high reward under $r$ at test time. There are two broad ways to influence the outputs of a pre-trained model:

**(i) Fine-tuning**. Techniques such as reinforcement learning from human feedback (RLHF) fine-tune the model so that high-reward responses are more likely (Christiano et al., 2017; Ouyang et al., 2022).

**(ii) Test-time steering**. Instead of additional training, these methods rely purely on how inference is conducted. Test-time steering treats compute at generation time as the resource to be allocated, biasing outputs toward higher-reward responses by modifying the decoding process (Welleck et al., 2024).

A simple and widely studied test-time approach is Best-of-$N$ sampling (Nakano et al., 2021; Touvron et al., 2023). For a given prompt $x$, we generate $N$ candidate responses $y_1, \dots, y_N \sim \pi(x)$, evaluate each with $r(x, y_i)$, and select the response with the highest score. Best-of-$N$ has been shown to substantially improve output quality across tasks, both empirically and theoretically, (Wang et al., 2025; Sun et al., 2024; Beirami et al., 2024; Yang et al., 2024), but it can be computationally expensive since it requires $N$ forward passes per prompt, with $N$ fixed in advance. Moreover, in practice, $N$ is typically fixed. However, some prompts may yield strong candidates after only a few samples, while others require more, making a uniform budget potentially wasteful. This motivates the design of adaptive test-time strategies that allocate compute more efficiently across prompts.

## 2.3 OPTIMAL STOPPING, PANDORA'S BOX, AND WEITZMAN'S ALGORITHM

The Pandora's Box problem (Weitzman, 1978) is a classical optimal stopping problem: a decision-maker faces $k$ boxes, each box $i \in [k]$ containing a reward drawn from a known distribution $D_i$ and requiring an opening cost $c_i > 0$. The objective is to adaptively decide which boxes to open and when to stop, maximizing the best observed reward minus total costs. Once opened, a box's reward can be claimed at any time.

This framework maps naturally to LLM generation. For a prompt $x$, an LLM $\pi$ samples responses from $\pi(x)$, with rewards assigned by a deterministic model $r$, inducing a reward distribution $D_{r,\pi(x)}$. Each generation incurs a computational cost $c$ that depends on the prompt, model, and reward function. Thus, generating responses mirrors opening boxes: each sample reveals a reward at cost $c$, and the decision-maker must trade off computation against reward.

A particularly relevant special case is when all boxes share the same distribution $D$ and cost $c$, modeling repeated queries to a single LLM. We focus on this setting throughout, though extensions to multiple LLMs are natural directions for future work.

A fundamental concept in solving this problem is the fair-cap value, the threshold where expected excess reward exactly covers the opening cost.

**Definition 1** (Fair-cap value). *Let $D$ be a distribution and $c > 0$ be the cost value for each sample. The fair-cap value $\tau \in \mathbb{R}$ associated with $(D, c)$, denoted $\tau(D, c)$, is the number satisfying the equality $\mathbb{E}_{v \sim D}[[v - \tau]_+] = c$, where $[\cdot]_+ = \max(0, \cdot)$.*

Weitzman's celebrated algorithm provides the optimal stopping strategy using fair-cap values when distributions are known. Though Weitzman's algorithm is defined for an arbitrary collection of not-necessarily-identical distributions in general, we describe its special case for infinitely-many boxes with identical distributions below. This corresponds to our focus on a single LLM which can be queried an unbounded number of times.

**Definition 2** (Weitzman's Algorithm for infinitely-many identical boxes). *Let $D$ be a known distribution, $c > 0$ be the cost value for each sample, and $\tau := \tau(D, c)$ be the fair-cap value. The algorithm samples from $D$ until it observes a report exceeding $\tau$. More formally: Letting $v_1, v_2, \dots \sim D^\infty$ be a countably infinite sequence of i.i.d. rewards, the stopping time of Weitzman's algorithm is the random variable $T_W := \inf\{n \geq 1 : \max_{i \leq n} v_i \geq \tau\}$.*

In practice, reward distributions are effectively unknown. While implicitly encoded in the model weights, they lack compact representation and are only accessible through sampling. The meta-generation problem thus becomes a Pandora's Box problem with an *unknown* single distribution but known cost value. To evaluate algorithms in the *unknown* distribution setting, we compare against an oracle that knows the true distribution and executes Weitzman's algorithm optimally.

**Definition 3** (Additive Sub-optimality Gap). *Consider distribution $D$ with cost $c$. Let $W$ denote Weitzman's optimal policy with full knowledge of $D$, achieving net payoff $R_W$ (maximum reward minus total costs). For any policy $S$ that learns $D$ only through sampling, with payoff $R_S$, the additive sub-optimality gap is $\mathbb{E}_D[R_W - R_S]$.*

In general, if the reward distribution is allowed to be picked completely adversarially, then there is no hope for designing a single, minimax optimal stopping policy $S$ whose additive sub-optimality gap is uniformly bounded across all distributions. As a result, we will assume that we have a *known* distribution family $\mathcal{F}$ such that the *unknown* $D \in \mathcal{F}$.

# 3 PANDORA'S BOX WITH UNKNOWN REWARD DISTRIBUTIONS

When the distribution $D$ is unknown, the fair-cap value $\tau$ must be learned from data. Our main algorithm, *UCB Pandora's Box*, adapts the upper confidence bound (UCB) principle from multi-armed-bandit theory (Auer et al., 2002). The algorithm iteratively samples rewards and uses them along with the family $\mathcal{F}$ to construct an upper confidence bound $\tau^+$ on the fair-cap value $\tau$. It stops once the maximum observed reward $M$ exceeds the UCB on $\tau$. Pseudo-code is given in Algorithm 2.

The specific update for the UCB $\tau^+$ depends on $\mathcal{F}$ and the method for constructing confidence bounds for $\tau$. In particular, $\mathcal{F}$ must be "nice" enough to admit an *anytime-valid confidence sequence* for $\tau$. We define this rigorously and provide one such example in Section 3.1. In practice, the confidence parameter is a hyperparameter that influences the exploration-exploitation balance.

## 3.1 MAIN THEORETICAL RESULT

As previously highlighted, UCB Pandora's Box requires an anytime-valid upper confidence bound on the fair-cap value $\tau$. Definition 4 makes this precise.

**Definition 4** (Anytime Valid Upper Confidence Bound on the Fair-cap Value). *Let $\mathcal{F}$ be a family of distributions and $c > 0$ be the cost value for each sample. A function $\tau^+ : \mathbb{N} \times (0,1) \times \mathbb{R}^\star \to \mathbb{R}$ is an* anytime valid upper confidence bound *of the fair-cap value with* width function $\sigma : \mathbb{N} \times (0,1) \times \mathbb{R}$ *for $\mathcal{F}$ if for every $\delta \in (0,1)$ and $D \in \mathcal{F}$, we have*

$$\mathbb{P}_{v_{1:\infty} \sim D^\infty} \left[ \forall n \in \mathbb{N} : \tau \in [\tau_\delta^+(n, v_{1:n}) - \sigma_{\delta,\tau}(n), \tau_\delta^+(n, v_{1:n})] \right] \geq 1 - \delta.$$

*where $\tau = \tau(D, c)$ is the fair-cap value.*

If $\mathcal{F}$ is a parametric family of distributions, the fair-cap value often admits a simple monotonic dependence on the distribution's parameters. This observation allows us to obtain an upper confidence bound on the fair-cap value by applying the same monotonic transformation to an upper confidence bound on the parameters themselves. For example, in Section 6 we show that when $\mathcal{F}$ is the class of Exponential distributions, an upper confidence bound on the mean directly yields an upper confidence bound on the fair-cap value. More generally, constructing a confidence sequence for $\tau$ can be reduced to two steps: (1) build a confidence sequence for the distribution's parameters, and (2) propagate it through the monotonic mapping to obtain a confidence sequence for the fair-cap value.

We now present Theorem 5, our main theoretical result of this section, which upper bounds the additive sub-optimality gap of UCB Pandora's Box algorithm.

**Theorem 5** (Upper bound on Additive Sub-optimality). *Let $\mathcal{F}$ be a family of distributions and $c > 0$ be the cost value for each sample. Let $\tau_\delta^+(n, v_{1:n})$ be an anytime upper confidence bound with deterministic width $\sigma_{\delta,\tau}(n)$ for $\mathcal{F}$ according to Definition 4. i.e., for every distribution $D \in \mathcal{F}$, on the event $E_\delta := \left\{ \forall n \geq 1 : \tau \leq \tau_\delta^+(n, v_{1:n}) \leq \tau + \sigma_{\delta,\tau}(n) \right\}$ we have that $\mathbb{P}_{v_1, v_2, \cdots \sim D}(E_\delta) \geq 1 - \delta$. Consider the two stopping policies:*

- ***Weitzman policy:*** $T_W := \inf\{n \geq 1 : \max_{i \leq n} v_i \geq \tau\}$ *w/* $R_W := \max_{i \leq T_W} v_i - c\,T_W$.

- **UCB policy:** $T_U := \inf\{n \geq 1 : \max_{i \leq n} v_i \geq \tau_\delta^+(n, v_{1:n})\}$ w/ $R_U := \max_{i \leq T_U} v_i - c\, T_U$.

*Then for every $D \in \mathcal{F}$ and $\delta \in (0,1)$, we have that*

$$\mathbb{E}_D\left[(R_W - R_U)\mathbf{1}_{E_\delta}\right] \leq \sum_{n=1}^{\infty} \sigma_{\delta,\tau}(n)\,(1 - F_D(\tau))\Big(F_D^{n-1}\big(\tau + \sigma_{\delta,\tau}(n)\big) - F_D^{n-1}(\tau)\Big),$$

*where $F_D$ is the* CDF *of $D$ and $\tau = \tau(D,c)$ is the fair-cap value of $D$.*

The proof of Theorem 5 can be found in Appendix E.

## 3.2 Example: Exponential distribution with unknown parameter $\lambda$

The upper bound in Theorem 5 is abstract and instance-dependent. To obtain a more concrete result, we instantiate it with the family $\mathcal{F}$ of Exponential distributions parametrized by $\lambda \in (0, 1/(ce)]$.

For $\lambda$ in this range, let $D_\lambda \in \mathcal{F}$ denote the Exponential distribution with rate $\lambda$, with CDF $F_\lambda(x) = 1 - e^{-\lambda x}$ and PDF $f_\lambda(x) = \lambda e^{-\lambda x}$. The restriction $\frac{1}{\lambda} \geq ec$ ensures that the sampling cost $c$ is at most $1/e$ of the expected reward. Indeed, if $c > 1/\lambda$, not even a single sample would be worthwhile.

Theorem 6 provides an any-time valid UCB on the fair-cap value for distributions in $\mathcal{F}$.

**Theorem 6** (Anytime-valid Upper Confidence Bound for Exponential Fair-cap value)**.** *Let $c > 0$ be the sampling cost and $\mathcal{F}$ be the class of Exponential distributions with parameter $\lambda \in (0, \frac{1}{c \cdot e}]$. Then, the function $\tau_\delta^+(n, v_{1:n}) = \widehat{\mu}_n(1 + r_\delta(n)) \log\left(\frac{\widehat{\mu}_n(1 + r_\delta(n))}{c}\right)$ with width function $\sigma_{\delta,\tau}(n) = 16\tau \cdot r_\delta(n)$ is an* anytime valid upper confidence bound *on the fair-cap value, where $\widehat{\mu}_n = \frac{1}{n}\sum_{i=1}^n v_i$, and $r_\delta(n) := \min\left\{\frac{1}{2}, \sqrt{\frac{6}{n}\log\left(\frac{2n(n+1)}{\delta}\right)}\right\}$.*

The proof of Theorem 6 (Appendix E.2) proceeds by expressing the fair-cap value of an Exponential distribution as a monotonic function of its mean, constructing an anytime-valid UCB for the mean, and then transferring this bound to the fair-cap value. With Theorem 6 in hand, we can now explicitly compute the right-hand side of the expected sub-optimality gap in Theorem 5.

**Corollary 7** (Sub-optimality gap upper bound for Exponential distribution)**.** *Let $c > 0$ be the sampling cost and $\mathcal{F}$ be the class of Exponential distribution with parameter $\lambda \in (0, \frac{1}{c \cdot e}]$. Then, for every $\delta \in (0,1)$ and $D_\lambda \in \mathcal{F}$ we have that*

$$\sum_{n=1}^{\infty} \sigma_{\delta,\tau}(n) \cdot (1 - F_\lambda(\tau)) \cdot (F_\lambda^{n-1}(\tau + \sigma_{\delta,\tau}(n)) - F_\lambda^{n-1}(\tau)) \leq \widetilde{O}_\delta\left(\frac{1}{\lambda}\right),$$

*where $\sigma_{\delta,\tau}(n)$ is defined as in Theorem 6, $\tau = \tau(D_\lambda, c)$ is the fair-cap value, and $\widetilde{O}_\delta(\cdot)$ hides polylog terms of $\frac{1}{\delta}$. As a result, we have that for every $\delta \in (0,1)$ and $D_\lambda \in \mathcal{F}$ the additive sub-optimality gap for* UCB *Pandora's Algorithm satisfies $\mathbb{E}_{D_\lambda}\left[(R_W - R_U)\mathbf{1}_{E_\delta}\right] \leq \widetilde{O}_\delta\left(\frac{1}{\lambda}\right).$*

At a high level, Corollary 7 (proved in Appendix E.2) shows that under event $E_\delta$ (see Theorem 5), the expected sub-optimality gap of UCB Pandora's Box is bounded by the mean $1/\lambda$ (up to logarithmic factors in $\delta$). In Section 4, we leverage these results to develop prompt-adaptive inference-time optimization methods.

## 4 Applying UCB Pandora's Box to Inference-time Optimization

Building on the connection between test-time steering and the Pandora's Box problem established in Section 2.3, we now cast inference-time optimization as an instance of the Pandora's Box problem. Each generation from the base LLM can be viewed as opening a costly box whose payoff is the reward assigned by $r$. The challenge is to decide when to stop sampling: too early risks missing high-reward outputs, while too late wastes computation. In what follows, we develop adaptive stopping rules that navigate this tradeoff without prior knowledge of the underlying reward distribution.

## 4.1 THE REWARD SCALING CHALLENGE

A key subtlety in framing inference-time optimization as a Pandora's Box problem is deciding what makes a response "high quality." Using a fixed reward threshold is inadequate because reward scales vary dramatically across prompts. For example, Figure 4 in Appendix D shows large variance in median rewards across 100 prompts, even when evaluated with the same model.

To enable cost-shared semantics across prompts with different reward scales, we adopt a percentile-based approach. Specifically, we select a gold standard percentile $\alpha$ of the reward distribution as our quality benchmark. This choice naturally adapts to each prompt's reward scale while maintaining consistent quality standards. Throughout our work, we set $\alpha = 0.99$ to compete with best-of-$N$ sampling (where typically $\alpha \approx 1 - \frac{1}{N}$), though practitioners may adjust this parameter based on quality requirements. We formalize this approach through what we term the "acceptance criterion":

**Definition 8** (Acceptance Criterion). *A response $y \in \mathcal{Y}$ to prompt $x \in \mathcal{X}$ is* acceptable *with respect to the LM-reward model pair $(\pi, r)$ if its reward $r(x, y)$ exceeds the $\alpha$-percentile of the reward distribution $D_{r,\pi(x)}$, denoted $D^\alpha_{r,\pi(x)}$. We set $\alpha = 99$ throughout, though larger values correspond to stricter acceptance.*

Even so, scaling issues remain. For two prompts $x_1$ and $x_2$, the 99th percentiles $D^\alpha_{r,\pi(x_1)}$ and $D^\alpha_{r,\pi(x_2)}$ may differ greatly (e.g., $D^\alpha_{r,\pi(x_1)} \ll D^\alpha_{r,\pi(x_2)}$), yet exceeding either yields the same utility $B$. This mismatch between reward magnitude and utility motivates the need for *normalization*.

## 4.2 BRADLEY-TERRY TRANSFORMATION FOR REWARD NORMALIZATION

We resolve the scaling issue through a transformation inspired by the Bradley–Terry model, which also underlies RLHF training (Bradley & Terry, 1952; Christiano et al., 2017). In RLHF, the probability of preferring response $A$ over $B$ is modeled as $\frac{e^{r_A}}{e^{r_A}+e^{r_B}}$, where $r_A$ and $r_B$ are their rewards. We adapt this idea by comparing each response against an acceptance threshold.

**Definition 9** (Acceptance Rate). *The* acceptance rate *of a response $y$ with reward $v_y = r(x, y)$ with respect to threshold $\kappa$ is defined as $\mathrm{AR}_\kappa(v) = \min\left\{2 \cdot \frac{e^v}{e^v+e^\kappa}, 1\right\}$.*

This transformation maps rewards into $[0, 1]$. The Bradley–Terry term $\frac{e^v}{e^v+e^\kappa}$ represents the probability of preferring a response with reward $v$ over one with reward $\kappa$. Intuitively, the acceptance rate approximates the probability that an end-user accepts the response: below-threshold responses are accepted in proportion to their quality relative to $\kappa$, while at- or above-threshold responses are accepted with certainty. This normalization enables consistent comparisons across prompts with different reward scales. In our application, we set $\kappa = D^\alpha_{r,\pi(x)}$ as in Definition 8, making acceptance rates prompt-, model-, and reward-dependent. However, as the true distribution is unknown, we estimate $D^\alpha_{r,\pi(x)}$ from samples, which proves sufficiently accurate.

The acceptance rate naturally induces a utility function.

**Definition 10** (Utility Function). *For an acceptance threshold $\kappa$, the utility function $u_\kappa : \mathbb{R} \to [0, B]$ maps rewards to* utilities *via $u_\kappa(v) = B \cdot \mathrm{AR}_\kappa(v)$, where $B > 0$ is the maximum achievable utility.*

Here, responses below the threshold map to utilities in $[0, B)$, while acceptable responses map exactly to $B$. When $\kappa = D^\alpha_{r,\pi(x)}$, $\mathrm{AR}_\kappa(v)$ is the probability that a response with reward $v$ is accepted, $B$ is the utility of an accepted response, and hence $u_\kappa(v)$ is the *expected* utility of a response with reward $v$ (assuming rejected responses get no utility). Pushing the reward distribution $D_{r,\pi(x)}$ through $u_\kappa$ yields the utility distribution $U_{\kappa,r,\pi(x)} := u_\kappa(D_{r,\pi(x)})$ supported on $[0, B]$. Setting cost $c \in [0, B]$ then allows direct comparison of generation costs with achievable utilities.

## 4.3 THE ADAPTIVE ALGORITHM

We are now ready to adapt the UCB Pandora's Box Algorithm from Section 3.1 to the Best-of-$N$ inference-time optimization setting. Given a prompt $x \in \mathcal{X}$, LLM $\pi$, and reward model $r$, our algorithm sequentially generates responses and adaptively decides when to stop. After collecting a minimum number of samples, it estimates the reward distribution's tail. To do so, we exponentiate

the rewards and fit a shifted exponential distribution to the right-tail values (those above the median). From this fit, we construct both upper and lower confidence bounds (UCB/LCB) on the scale of the exponential distribution. The LCB on the scale is then used to derive a conservative estimate of the $\alpha$ percentile of the exponentiated rewards, while the UCB is used to bound the tail distribution itself. This combination yields an upper confidence bound on the fair-cap value of the true utility distribution. If the utility of the best reward observed so far exceeds this fair cap, the algorithm terminates; otherwise, it continues sampling. Pseudocode is given in Algorithm 1.

---

**Algorithm 1** Adaptive Best-of-N Sampling via UCB Pandora's Box

---

**Parameters:** Cost $c$, Max utility $B$, Minimum samples $t$, Percentile $\alpha$, Confidence parameter $\delta$.
**Input:** LLM $\pi$, reward model $r$, prompt $x$

1: Initialize: $S = \emptyset$ (observed rewards) and $M = -\infty$ (max reward)
2: **while** True **do**
3:     Generate response $y \sim \pi(x)$ and compute its reward $r_y = r(x, y)$.
4:     Update $M \leftarrow \max\{M, r_y\}$ and $S \leftarrow S \cup \{r_y\}$.
5:     **if** $|S| \geq t$ **then**
6:         **—Right Tail Estimation via Exponential Distribution—**
7:         Estimate tail shift $\widehat{\theta} \leftarrow e^{\text{median}(S)}$ and scale of shifted tail $\widehat{\mu} \leftarrow \text{mean}(\{e^r - \theta : r \in S \text{ such that } r > \text{median}(S)\})$.
8:         Apply upper and lower confidence estimation:

$$\mu_{\text{ucb}} \leftarrow \mu \cdot \left( 1 + \sqrt{\frac{\log |S| \cdot \log(1/\delta)}{|S|}} \right) \qquad \mu_{\text{lcb}} \leftarrow \mu \cdot \left( 1 - \sqrt{\frac{\log |S| \cdot \log(1/\delta)}{|S|}} \right).$$

9:         Let $\widehat{D}^{\text{ucb}} \leftarrow \text{ShiftedExp}(\theta, \frac{1}{\mu_{\text{ucb}}})$ and $\widehat{D}^{\text{lcb}} \leftarrow \text{ShiftedExp}(\theta, \frac{1}{\mu_{\text{lcb}}})$ be shifted exponential distributions with shift parameter $\theta$ and scales $\frac{1}{\mu_{\text{ucb}}}$ and $\frac{1}{\mu_{\text{lcb}}}$ respectively.
10:         **—Utility Transformation and Fair Cap Computation—**
11:         Let $\widehat{D}^{\alpha}_{r,\pi(x)} \leftarrow \log(\text{Percentile}_\alpha(\widehat{D}^{\text{lcb}}))$ be a LCB estimate of $D^{\alpha}_{r,\pi(x)}$.
12:         Define utility distribution $\widehat{U} \leftarrow u_\kappa(\log(\widehat{D}^{\text{ucb}}))$ where $\kappa = \widehat{D}^{\alpha}_{r,\pi(x)}$.
13:         Compute fair-cap value $\tau$ for $(\widehat{U}, c)$.
14:         **if** $u_\kappa(M) \geq \tau$ **then**
15:             **break**
16:         **end if**
17:     **end if**
18: **end while**
19: **Return** the response with reward $M$

---

**Implementation Efficiency.** The algorithm can be implemented with negligible overhead relative to generation costs. A priority queue maintains the median and tail statistics with $O(\log n)$ updates and $O(1)$ queries, while streaming updates eliminate redundant computation. Key distributional operations are closed-form. For example, the $\alpha$-percentile of the shifted exponential distribution is computed in $O(1)$ time from an analytical formula. Fair-cap computation requires solving $\mathbb{E}[\max(v - \tau, 0)] = c$ for the threshold $\tau$. We approximate the expectation via a Riemann sum with $\sim$5000 intervals. Empirically, this achieves under 1% relative error. The subroutine executes over 100 times per second[2], enabling real-time adaptation during generation. In practice, the overhead of adaptive stopping is negligible as LLM and reward model forward passes dominate runtime.

## 4.4 TARGET ACCEPTANCE RATE VARIANT

Algorithm 1 requires specifying the utility $B$ and cost $c$, which may be difficult to estimate in practice. To address this, we provide an alternative formulation that instead targets a desired acceptance rate. Rather than computing the fair-cap value from utility–cost tradeoffs, this variant sets a target acceptance rate $\tau_{\text{target}} \in [0, 1]$ that encodes the desired quality level relative to the acceptance threshold. For example, $\tau_{\text{target}} = 0.9$ seeks responses nearly as good as acceptable ones, while $\tau_{\text{target}} = 1$

---

[2]Measured on a single core of an AMD EPYC 7513 32-Core Processor.

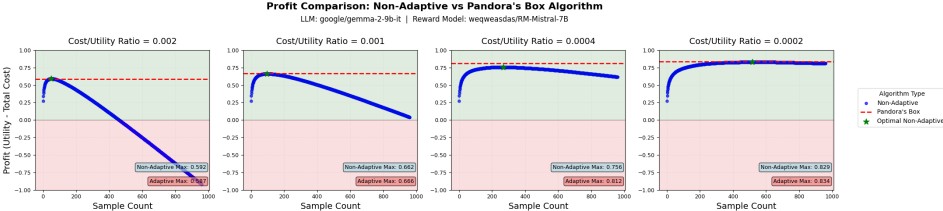

Figure 1: Our algorithm (red) matches optimal non-adaptive performance across varying cost ratios.

requires fully acceptable responses. The algorithm proceeds identically to Algorithm 1 except that the stopping condition is now fixed: it halts when $\mathrm{AR}_{\widehat{D}^{\alpha}_{r,\pi(x)}}(M) \geq \tau_{\text{target}}$[3]. This formulation is useful in settings where quality requirements are clear but utilities are hard to quantify, for instance, when "good enough" responses are well-defined but the value of marginal improvements is ambiguous.

## 5 EXPERIMENTAL RESULTS

### 5.1 EXPERIMENTAL SETUP

We evaluate our adaptive Best-of-$N$ algorithm on 100 prompts from AlpacaFarm (Dubois et al., 2023) and 100 from HH-RLHF (Bai et al., 2022). We use four LLM (Google-Gemma-2 (9B), Meta-Llama-3.1 (8B), Mistral-Instruct-v0.3 (7B), Qwen-2.5 (7B)) and two reward models (FsfairX-LLaMA3-RM-v0.1, RM-Mistral-7B; Dong et al., 2023; Xiong et al., 2024). This yields 1,600 generation profiles (2 datasets × 100 prompts × 4 LLMs × 2 reward models). For each profile, we generate 960 responses, compute rewards, and randomize response–reward orderings 100 times to remove ordering effects. We always fix the max utility $B = 1$, and only vary the cost $c$.

We benchmark against non-adaptive Best-of-$N$ across three tasks: (1) *profit optimization*, comparing utility–cost tradeoffs; (2) *win rate analysis*, under fixed compute budgets; and (3) *efficiency gains*, measuring compute savings at target quality levels. More experimental results are in Appendix F.

### 5.2 EXPERIMENT 1: PROFIT OPTIMIZATION

We evaluate how well our adaptive algorithm maximizes profit (utility minus total cost) relative to the best non-adaptive strategy. We consider four cost-to-utility ratios: $0.002, 0.001, 0.0004, 0.0002$. Figure 1 reports results for Mistral-Instruct-v0.3 (7B) with the RM-Mistral-7B reward model. The adaptive algorithm (red) either closely approximates or outperforms the profit envelope defined by the best non-adaptive strategies (blue/green), automatically achieving this performance without knowing the optimal $N$ in advance. Across all 1,600 generation profiles, the adaptive method outperforms the generator-dependent best non-adaptive algorithm in nearly all cases, demonstrating clear superiority (Figure 5).

### 5.3 EXPERIMENT 2: WIN RATE UNDER FIXED BUDGET

We compare our adaptive algorithm with non-adaptive Best-of-$N$ under equal computation budgets. For each prompt $x$, LLM $\pi$, and cost $c \in [10^{-5}, 10^{-3}]$, we: (1) run the adaptive algorithm on 100 random orderings and record the average sample count $\overline{n}_{\pi,x,c}$, (2) run non-adaptive Best-of-$N$ with $N = \overline{n}_{\pi,x,c}$ on the same orderings, and (3) compare maximum rewards, awarding half credit for ties. Figure 2 shows the results. The adaptive algorithm consistently outperforms non-adaptive Best-of-$N$, with win rates exceeding 54% across most cost settings. Gains are largest when costs are low (permitting more samples) or budgets are higher, where adaptive stopping better exploits variation across prompts. Figure 6 confirms these results across different datasets and reward models.

---

[3]In this variant, we estimate $\widehat{D}^{\alpha}_{r,\pi}(x)$ using $\widehat{D}^{\text{ucb}}$. Otherwise, the algorithm tends to stop prematurely by overestimating the acceptance rate of the maximum sample.

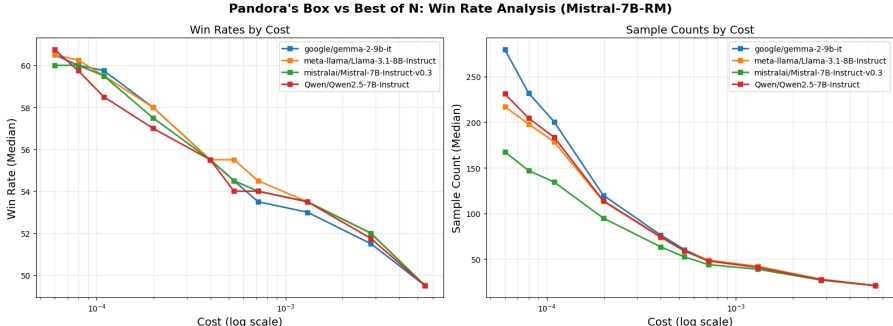

Figure 2: Win rate of the adaptive algorithm compared to non-adaptive Best-of-$N$. Adaptive stopping leverages computation more effectively, particularly at lower costs.

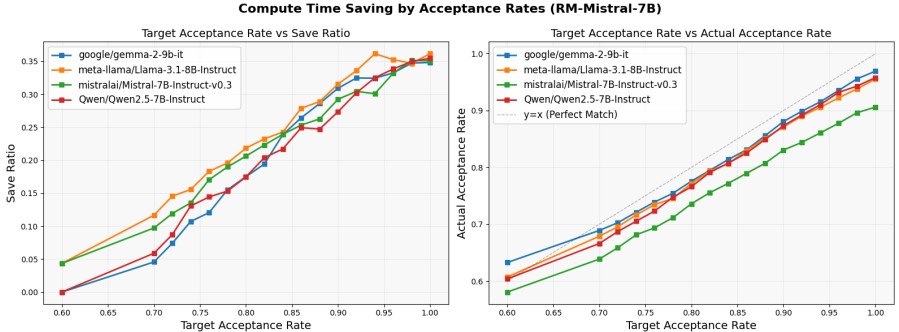

Figure 3: Adaptive algorithm achieves specified target acceptance rates while saving 15–35% of samples compared to non-adaptive Best-of-$N$.

### 5.4 EXPERIMENT 3: EFFICIENCY AT TARGET QUALITY LEVELS

We next evaluate the target acceptance rate variant, which allows users to directly specify a desired quality level. For target rates $\tau \in [0.60, 1]$ and each configuration (LLM $\pi$, prompt $x$, target $\tau$), we: (1) measure the adaptive algorithm's average acceptance rate $\overline{a}_{\pi,x,\tau}$ and sample count $\overline{n}_{\pi,x,\tau}$; (2) identify the non-adaptive $N^\star$ that achieves the same $\overline{a}_{\pi,x,\tau}$; and (3) compute the efficiency gain

$$\text{SaveRatio}_{\pi,x,\tau} = \frac{N^\star - \overline{n}_{\pi,x,\tau}}{N^\star}.$$

Figure 3 shows that the adaptive algorithm both tracks the target quality (right) and yields substantial savings in sample counts (left). For acceptance rates $0.75+$, it consistently reduces sampling by 15–35% relative to non-adaptive methods. Savings increase monotonically with stricter targets, from $\sim$15% up to $\sim$35%, reflecting more effective use of learned tail information at higher quality levels.

## 6 DISCUSSION

We highlight several directions for future investigation. **(1) Multi-model inference:** Although we consider a single LLM in this work, the Pandora's Box framework extends naturally to ensembles of models, where each model corresponds to a box type with its own cost–quality profile. An open question is whether adaptive algorithms can automatically route queries across models, reducing the need for hand-designed cascades (Yue et al., 2023). **(2) Tree search and reasoning:** Approaches such as tree-of-thought (Yao et al., 2023) and Monte Carlo tree search (Feng et al., 2023) also involve sequential explore–exploit trade-offs at each decision point. Optimal stopping may help formalize when to expand or backtrack in such settings, potentially improving efficiency relative to existing heuristics.

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

## A  DISCLOSURE OF LLM USAGE

LLMs were used to aid and polish the writing throughout the paper. In addition, LLMs were used for retrieval and discovery to help write the Related Works section.

## B  OTHER RELATED WORKS

**Extreme Bandits.** Another related line of work is the *extreme bandit problem*, which optimizes the maximum observed reward rather than cumulative reward (Streeter & Smith; Cicirello & Smith; Bhatt et al., 2021). This is closely aligned with best-of-$N$ sampling, where one selects the highest-quality output among multiple candidates. However, most extreme bandit formulations assume repeated rounds of play, while inference requires efficient decision-making in a single prompt. Our framework adapts these insights to one-shot inference with explicit cost–quality tradeoffs.

**Cascading, Routing, and Hybrid Strategies.** Beyond stopping and sampling, researchers have explored *cascaded or routed inference* to reduce cost. Chen et al. (2024) escalate from cheaper to larger models only when necessary, using self-testing to decide. Mohammadshahi et al. (2024) learn to route prompts among multiple models, balancing cost and performance. These strategies complement our focus by showing that adaptive allocation can occur across models as well as within a single model's sampling process.

## C  UCB PANDORA'S BOX ALGORITHM

Algorithm 2 provides the exact pseudo-code for the algorithm outlined in Section 3.

---
**Algorithm 2** UCB Pandora's Box Algorithm

---
**Input:** Distribution family $\mathcal{F}$, sampling cost $c$.
**Parameter:** Confidence level parameter $\delta > 0$.
 1: Initialize $S = \emptyset$ (set of observed sample values).
 2: Initialize $m = -\infty$ (maximum value seen so far).
 3: Initialize $\tau^+ = M_0$ (a large initial upper confidence bound for the fair cap).
 4: **while** $M < \tau^+$ **do**
 5:     Query sample and obtain $v \sim D$.
 6:     $S \leftarrow S \cup \{v\}$.
 7:     $M \leftarrow \max\{M, v\}$.
 8:     Update $\tau^+$: Compute the UCB for the fair-cap value $\tau$ of $(D, c)$, based on $S$, $\mathcal{F}$, and the confidence parameter $\delta$.
 9: **end while**
10: **Return:** $M$.

---

## D  MISSING FIGURES

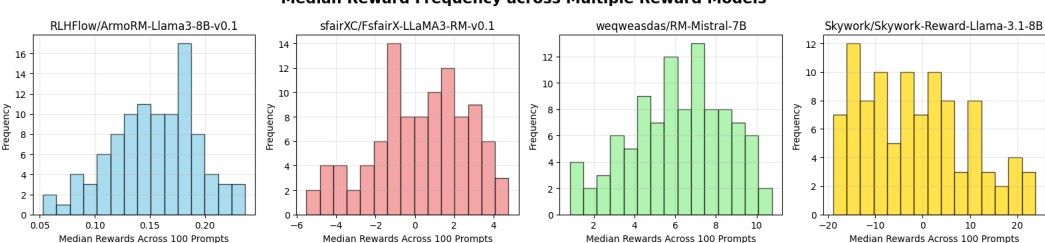

Figure 4: Median rewards across 960 generations for 100 prompts from the AlpacaEval dataset.

We find that the median reward across 960 generations can vary significantly across prompts.

# E  MISSING PROOFS AND THEORETICAL RESULTS

## E.1  PROOF OF THEOREM 5

In this section, we include all missing proofs and helper lemmas needed to prove Theorem 5. The following helper lemmas will be useful.

**Lemma 11.** *Let $D$ be any distribution, $c > 0$ be the cost value for each sample, and $\tau = \tau(D, c)$ be the fair-cap value according to Definition 1. For every $\sigma > 0$, we have that*

$$\mathbb{E}_{v \sim D}\left[[v - (\tau + \sigma)]_+\right] \geq c - \sigma \cdot (1 - F(\tau)),$$

*where $F$ denotes the CDF of $D$.*

*Proof.* Observe the following sequence of inequalities.

$$
\begin{aligned}
\mathbb{E}_{v \sim D}\left[[v - (\tau + \sigma)]_+\right] - c &= \mathbb{E}_{v \sim D}\left[[v - (\tau + \sigma)]_+\right] - \mathbb{E}_{v \sim D}\left[[v - \tau]_+\right] \\
&= \mathbb{P}_{v \sim D}[v \geq \tau]\, \mathbb{E}_{v \sim D}\left[[v - \tau - \sigma]_+ - (v - \tau)|v \geq \tau\right] \\
&\geq \mathbb{P}_{v \sim D}[v \geq \tau]\, \mathbb{E}_{v \sim D}\left[v - \tau - \sigma - (v - \tau)|v \geq \tau\right] \\
&= (1 - F(\tau)) \cdot (-\sigma).
\end{aligned}
$$

Rearranging completes the proof. $\qquad\square$

**Lemma 12.** *Let $D$ be any distribution, $c > 0$ be the cost value for each sample and $\tau = \tau(D, c)$ be the fair-cap value according to Definition 1. Then, for every $n \in \mathbb{N}$ and $\sigma > 0$ we have that*

$$\mathbb{P}_{v_{1:n-1} \sim D^{n-1}}\left[\max\{v_1, \ldots, v_{n-1}\} \in [\tau, \tau + \sigma]\right] \quad \leq \quad F^{n-1}(\tau + \sigma) - F^{n-1}(\tau).$$

*Proof.* Fix $n \geq 1$ and let $M_{n-1} = \max\{v_1, \ldots, v_{n-1}\}$. Observe that $\mathbb{P}[M_{n-1} \leq x] = F^{n-1}(x)$ where $F$ is the CDF of $D$ because $v_{1:n-1}$ are iid draws. Noting that

$$\mathbb{P}[M_{n-1} \in [\tau, \tau + \sigma]] = \mathbb{P}[M_{n-1} \leq \tau + \sigma] - \mathbb{P}[M_{n-1} \leq \tau]$$

completes the proof. $\qquad\square$

**Lemma 13.** *Let $D$ be any distribution, $c > 0$ be the cost value for each sample, and $\tau = \tau(D, c)$ be the fair-cap value according to Definition 1. Then, for every $n \in \mathbb{N}$ and $\sigma > 0$, we have that*

$$\mathbb{E}_{v_{1:n} \sim D^n}\left[\Delta_n \mathbf{1}_{B_n}\right] \geq -\sigma(1 - F(\tau))\mathbb{P}[B_n]$$

*where*

$$\Delta_n := [v_n - \max\{v_{1:n-1}\}]_+ - c,$$

*and $B_n$ is the event that*

$$\max\{v_{1:n-1}\} \in [\tau, \tau + \sigma].$$

*Proof.* (of Lemma 13) It suffices to prove the lower bound

$$\mathbb{E}[\Delta_n | v_{1:n-1}] \geq -\sigma(1 - F(\tau))$$

on the event that $B_n$ occurs. Since $v_n$ is independent of $v_{1:n-1}$, we have that

$$\mathbb{E}[\Delta_n | v_{1:n-1}] \geq \mathbb{E}\left[[v_n - (\tau + \sigma)]_+\right] - c \geq -\sigma \cdot (1 - F(\tau)),$$

where the last inequality follows by Lemma 11. $\qquad\square$

We are now ready to prove Theorem 5.

*Proof.* (of Theorem 5) Let $c > 0$ be the sampling cost and $\mathcal{F}$ be any family of distributions that admits an anytime-valid upper confidence bound $\tau_\delta^+(n, v_{1:n})$ with width function $\sigma_{\delta,\tau}(n)$. Fix some $D \in \mathcal{F}$ and consider the iid sequence $v_1, v_2 \cdots \sim D$. let $\tau = \tau(D, c)$ be the fair-cap value of $(D, c)$ and $F$ denote the CDF of $D$. For every $\delta \in (0, 1)$, let $E_\delta$ denote the event in Definition 4.

For $n \geq 1$, write $M_{n-1} = \max\{v_{1:n-1}\}$ and define

$$\Delta_n := [v_n - M_{n-1}]_+ - c$$

and

$$B_n := \{M_{n-1} \in [\tau, \tau + \sigma_{\delta,\tau}(n)]\}.$$

On the event $E_\delta$, the two policies disagree only when event $B_n$ occurs. Hence,

$$\mathbb{E}\left((R_W - R_U)\mathbf{1}_{E_\delta}\right) \leq \mathbb{E}\left[\sum_{n=1}^{\infty} -\Delta_n \mathbf{1}_{B_n} \mathbf{1}_{E_\delta}\right].$$

We aim to use Fubini's theorem to swap the expectation with the sum. To do so, we need to show that

$$\sum_{n=1}^{\infty} \mathbb{E}\left[|-\Delta_n \mathbf{1}_{B_n} \mathbf{1}_{E_\delta}|\right] < \infty.$$

To see this, note that

$$|-\Delta_n \mathbf{1}_{B_n} \mathbf{1}_{E_\delta}| \leq (c + [v_n - M_{n-1}]_+)\mathbf{1}_{B_n} \leq (c + [v_n - \tau]_+)\mathbf{1}_{B_n},$$

where the last inequality follows from the fact that on $B_n$, we have that $M_{n-1} \geq \tau$. Thus, it suffices to upper bound

$$\mathbb{E}\left[(c + [v_n - \tau]_+)\mathbf{1}_{B_n}\right].$$

By conditioning on $M_{n-1}$ and using the tower law, we can write

$$\begin{aligned}
\mathbb{E}\left[(c + [v_n - \tau]_+)\mathbf{1}_{B_n}\right] &= \mathbb{E}\left[\mathbb{E}\left[(c + [v_n - \tau]_+)\mathbf{1}_{B_n}|M_{n-1}\right]\right] \\
&= \mathbb{E}\left[\mathbb{E}\left[c + [v_n - \tau]_+|M_{n-1}\right]\mathbf{1}_{B_n}\right] \\
&= \mathbb{E}\left[\mathbb{E}\left[c + [v_n - \tau]_+\right]\mathbf{1}_{B_n}\right] \\
&= \mathbb{E}\left[2c\mathbf{1}_{B_n}\right] \\
&= 2c \cdot \mathbb{P}\left[B_n\right] \\
&= 2c \cdot \left(F^{n-1}(\tau + \sigma_{\delta,\tau}(n)) - F^{n-1}(\tau)\right) \\
&\leq 2c \cdot F^{n-1}(\tau + \sigma_{\delta,\tau}(n)),
\end{aligned}$$

where the third equality is by independence of $v_n$ and $M_{n-1}$, the fourth equality is by definition of $\tau$, and the sixth equality by Lemma 12. Hence,

$$\sum_{n=1}^{\infty} \mathbb{E}\left[|-\Delta_n \mathbf{1}_{B_n} \mathbf{1}_{E_\delta}|\right] \leq 2c \sum_{n=1}^{\infty} F^{n-1}(\tau + \sigma_{\delta,\tau}(n)) < \infty,$$

since $\sigma_{\delta,\tau}(n) \to 0$ as $n \to \infty$. Accordingly, by Fubini's, theorem, we have that

$$\mathbb{E}\left((R_W - R_U)\mathbf{1}_{E_\delta}\right) \leq \sum_{n=1}^{\infty} \mathbb{E}\left[-\Delta_n \mathbf{1}_{B_n} \mathbf{1}_{E_\delta}\right].$$

By Lemma 13 we have that

$$\sum_{n=1}^{\infty} \mathbb{E}\left[-\Delta_n \mathbf{1}_{B_n} \mathbf{1}_{E_\delta}\right] \leq \sum_{n=1}^{\infty} \sigma_{\delta,\tau}(n)(1 - F(\tau))\mathbb{P}\left[B_n\right].$$

Finally, by Lemma 12, we have that

$$\mathbb{P}\left[B_n\right] = F^{n-1}(\tau + \sigma_{\delta,\tau}(n)) - F^{n-1}(\tau),$$

which completes the proof. $\qquad\square$

### E.2 Proof of Corollary 7

In this section, we include all missing proofs and helper lemmas needed to prove Corollary 7.

In order to use Theorem 5, we need to specify several things. First, Lemma14 computes the fair cap value for an Exponential distribution with parameter $\lambda$.

**Lemma 14** (Fair-cap value for Exponential distribution). *Let $D_\lambda$ be the exponential distribution with parameter $\lambda > 0$. Then, for every $c > 0$, the fair cap value $\tau$ associated with $(D, c)$ is $\frac{\log\left(\frac{1}{\lambda c}\right)}{\lambda}$*

*Proof.* For $v \sim \mathrm{Exp}(\lambda)$, we have

$$\mathbb{E}\left[[v - \tau]_+\right] = \int_\tau^\infty (x - \tau)\lambda e^{-\lambda x} = \frac{e^{-\lambda\tau}}{\lambda}.$$

Setting this equal to $c > 0$ and solving for $\tau$ completes the proof. $\qquad\square$

Next, we need to derive an anytime valid upper confidence bound on the fair-cap value for the Exponential distribution with parameter $\lambda$. Note that an upper confidence bound on $1/\lambda$ gives an upper confidence bound on $\tau$ when $D_\lambda$ is an exponential distribution with parameter $\lambda$. Hence, as a first step, Lemma 15 derives an anytime-valid confidence sequence for the mean of the Exponential distribution.

**Lemma 15** (Anytime-valid Confidence Sequence for Exponential distribution). *Fix $\lambda > 0$ and let $D_\lambda$ be an Exponential distribution with parameter $\lambda$. Define*

$$r_\delta(n) := \min\left\{\frac{1}{2}, \sqrt{\frac{6}{n}\log\left(\frac{2n(n+1)}{\delta}\right)}\right\}.$$

*Then, for every $\delta \in (0, 1)$, we have that*

$$\mathbb{P}_{v_{1:\infty} \sim D^\infty}\left[\forall n \geq 1 : \mu \in [\widehat{\mu}_n(1 - r_\delta(n)), \widehat{\mu}_n(1 + r_\delta(n))]\right] \geq 1 - \delta$$

*where $\mu = \frac{1}{\lambda}$ and $\widehat{\mu}_n = \frac{1}{n}\sum_{i=1}^n v_i$.*

*Proof.* Let $\lambda > 0$ and $D_\lambda$ be an exponential distribution with parameter $\lambda$. Let $v_1, v_2, \dots$ denote a sequence of iid draws from $D_\lambda$. Define $w_i = v_i/\mu$, where $\mu = 1/\lambda$. Note that $w_i \sim \mathrm{Exp}(1)$. Fix some $n \geq 1$ and define $S_n := \sum_{i=1}^n w_i \sim \mathrm{Gamma}(n, 1)$. Note that $\frac{\widehat{\mu}_n}{\mu} = \frac{S_n}{n}$. For any $s \in (0, 1)$ we will show that

$$\mathbb{P}\left[\frac{\widehat{\mu}_n}{\mu} \geq \frac{1}{1-s}\right] \leq e^{-n\psi_+(s)}$$

and

$$\mathbb{P}\left[\frac{\widehat{\mu}_n}{\mu} \leq \frac{1}{1+s}\right] \leq e^{-n\psi_-(s)},$$

where

$$\psi_+(s) := \frac{s}{1-s} + \log(1-s),$$

and

$$\psi_-(s) := \log(1+s) - \frac{s}{1+s}.$$

Starting with the upper tail, by Markov's inequality and the MGF of the $\mathrm{Gamma}(n, 1)$, we have that

$$\mathbb{P}[S_n \geq a] = \mathbb{P}\left[e^{sS_n} \geq e^{sa}\right] \leq \frac{\mathbb{E}\left[e^{sS_n}\right]}{e^{sa}} = (1-s)^{-n}e^{-sa}.$$

With $a = \frac{n}{1-s}$, we have that

$$\mathbb{P}\left[\frac{\widehat{\mu}_n}{\mu} \geq \frac{1}{1-s}\right] = \mathbb{P}[S_n \geq a] \leq \exp\left(-n\left(\frac{s}{1-s} + \log(1-s)\right)\right) = e^{-n\psi_+(s)}.$$

For the lower tail, we use the fact that $x \to e^{-sx}$ is decreasing to get that

$$\mathbb{P}\left[S_n \leq a\right] = \mathbb{P}\left[e^{-sS_n} \geq e^{-sa}\right] \leq \frac{\mathbb{E}\left[e^{-sS_n}\right]}{e^{-sa}} = (1+s)^{-n}e^{sa}.$$

Using $a = \frac{n}{1+s}$, gives that

$$\mathbb{P}\left[\frac{\widehat{\mu}_n}{\mu} \leq \frac{1}{1+s}\right] \leq e^{-n\psi_-(s)}.$$

Now, we show that for any $s \in (0, 1/2)$, we have that $\psi_+(s) \geq \frac{s^2}{6}$ and $\psi_-(s) \geq \frac{s^2}{6}$. For $\psi_+$, define $h_+(s) = \psi_+(s) - s^2/6$. Then, note that

$$h'_+(s) = \frac{s}{(1-s)^2} - \frac{s}{3} = s\left(\frac{1}{(1-s)^2} - \frac{1}{3}\right) \geq \frac{2s}{3} \geq 0.$$

Since $h_+(0) = 0$ and $h'_+ \geq 0$ we have that $h_+(s) \geq 0$. For $\psi_-$, we have that

$$\psi_-(s) = \log(1+s) - \frac{s}{1+s} \geq \left(s - \frac{s^2}{2}\right) - \left(s - \frac{s^2}{1+s}\right) = s^2\left(\frac{1}{1+s} - \frac{1}{2}\right) = \frac{s^2(1-s)}{2(1+s)} \geq \frac{s^2}{6},$$

where the first inequality is due to the fact that $\log(1+s) \geq s - \frac{s^2}{2}$ for $s \in (0,1)$ and the last inequality uses the fact that $s \leq 1/2$. We now complete the proof by picking a summable sequence $\delta_n$. Namely, pick $\delta_n = \frac{\delta}{n(n+1)}$ and note that $\sum_{n\geq 1} \delta_n = \delta$. At time $n \geq 1$, pick

$$r_\delta(n) := \min\left\{\frac{1}{2}, \sqrt{\frac{6}{n}\log\left(\frac{2}{\delta_n}\right)}\right\} = \min\left\{\frac{1}{2}, \sqrt{\frac{6}{n}\log\left(\frac{2n(n+1)}{\delta}\right)}\right\}.$$

Our previous analysis, then gives that

$$\mathbb{P}\left[\mu \notin [\widehat{\mu}_n(1 - r_\delta(n)), \widehat{\mu}_n(1 + r_\delta(n))]\right] \leq \delta_n.$$

Hence, by the union bound, we have that

$$\mathbb{P}\left[\exists n \geq 1 : \mu \notin [\widehat{\mu}_n(1 - r_\delta(n)), \widehat{\mu}_n(1 + r_\delta(n))]\right] \leq \sum_{n=1}^{\infty} \delta_n = \delta,$$

which completes the proof. $\qquad\square$

Note that the anytime-valid upper confidence bound on the mean $1/\lambda$ given by Theorem 15 can be computed just from the observed rewards. Hence, it applies to *every* distribution in our distribution family $\mathcal{F}$. With Lemma 15 in hand, we can now prove Theorem 6, which gives an anytime-valid upper confidence bound on the fair-cap value for the family of Exponential distributions.

*Proof.* (of Theorem 6) Let $c > 0$ be the sampling cost and $\mathcal{F}$ be the class of Exponential distribution with parameter $\lambda \in (0, \frac{1}{c\cdot e}]$. Fix a distribution $D_\lambda \in \mathcal{F}$. Let $\tau = \tau(D, c)$ be the fair-cap value and $\mu = \frac{1}{\lambda}$ be the mean of $D_\lambda$. Finally, let $\delta \in (0,1)$ and consider the iid sequence $v_1, v_2, \cdots \sim D_\lambda$.

By Lemma 15, we have that

$$\mathbb{P}\left[\forall n \geq 1 : \mu \in [\widehat{\mu}_n(1 - r_\delta(n)), \widehat{\mu}_n(1 + r_\delta(n))]\right] \geq 1 - \delta.$$

From Lemma14, we know that for the exponential distribution, its cap value is a monotonic in $\frac{1}{\lambda}$. Hence, with probability $1 - \delta$, we have that

$$\mathbb{P}\left[\forall n \geq 1 : \tau \in \left[\widehat{\mu}_n(1 - r_\delta(n))\log\left(\frac{\widehat{\mu}_n(1 - r_\delta(n))}{c}\right), \widehat{\mu}_n(1 + r_\delta(n))\log\left(\frac{\widehat{\mu}_n(1 + r_\delta(n))}{c}\right)\right]\right] \geq 1 - \delta.$$

Take $\tau_\delta^+(n, v_{1:n}) := \widehat{\mu}_n(1 + r_\delta(n))\log\left(\frac{\widehat{\mu}_n(1 + r_\delta(n))}{c}\right)$. To complete the proof, it suffices to upper bound the difference

$$\widehat{\mu}_n(1 + r_\delta(n))\log\left(\frac{\widehat{\mu}_n(1 + r_\delta(n))}{c}\right) - \widehat{\mu}_n(1 - r_\delta(n))\log\left(\frac{\widehat{\mu}_n(1 - r_\delta(n))}{c}\right).$$

Consider the function $g(\mu) = \mu \log(\mu/c)$. Then, by the Mean Value Theorem and the fact that $g'(\mu) = 1 + \log\left(\frac{\mu}{c}\right)$, we have that

$$g(\widehat{\mu}_n(1 + r_\delta(n))) - g(\widehat{\mu}_n(1 - r_\delta(n))) \leq \left(1 + \log\left(\frac{\widehat{\mu}_n(1 + r_\delta(n))}{c}\right)\right)(2\widehat{\mu}_n r_\delta(n))$$

$$\leq \left(1 + \log\left(\frac{\widehat{\mu}_n(1 + r_\delta(n))}{c}\right)\right)\left(\frac{2r_\delta(n)\mu}{1 - r_\delta(n)}\right)$$

$$\leq \left(1 + \log\left(\frac{\mu(1 + r_\delta(n))}{c(1 - r_\delta(n))}\right)\right)\left(\frac{2r_\delta(n)\mu}{1 - r_\delta(n)}\right).$$

Recall that $r_\delta(n) \leq \frac{1}{2}$ by definition. Hence,

$$\left(1 + \log\left(\frac{\mu(1 + r_\delta(n))}{c(1 - r_\delta(n))}\right)\right)\left(\frac{2r_\delta(n)\mu}{1 - r_\delta(n)}\right) \leq \left(3 + \log\left(\frac{\mu}{c}\right)\right)(4r_\delta(n)\mu)$$

$$= 4r_\delta(n) \cdot \tau \cdot \left(1 + \frac{3}{\log\left(\frac{\mu}{c}\right)}\right)$$

$$\leq 16r_\delta(n) \cdot \tau,$$

where the last inequality follows from the assumption that $\mu \geq e \cdot c$. Hence, altogether, we can take

$$\sigma_{\delta,\tau}(n) = 16r_\delta(n) \cdot \tau,$$

which completes the proof. $\qquad\square$

Finally, we use Theorem 5 to bound the sub-optimality gap for UCB Pandora's Box algorithm for the family $\mathcal{F}$ of Exponential distributions.

*Proof.* (of Corollary 7) Let $\mathcal{F}$ be the class of Exponential distribution with parameter $\lambda \in (0, \frac{1}{c \cdot e}]$. Fix a distribution $D_\lambda \in \mathcal{F}$. Let $\tau = \tau(D, c)$ be the fair-cap value and let $\delta \in (0, 1)$.

Define

$$S := \sum_{n=1}^{\infty} \sigma_{\delta,\tau}(n)(1 - F_\lambda(\tau)) \cdot (F_\lambda^{n-1}(\tau + \sigma_{\delta,\tau}(n)) - F_\lambda^{n-1}(\tau))$$

By the Mean Value Theorem and the fact that the PDF $f$ is monotonically decreasing, we have that for some $b \in (\tau, \tau + \sigma_{\delta,\tau}(n))$,

$$F_\lambda^{n-1}(\tau + \sigma_{\delta,\tau}(n)) - F_\lambda^{n-1}(\tau) = \sigma_{\delta,\tau}(n) \cdot (n-1) \cdot F_\lambda^{n-2}(b) \cdot f_\lambda(b)$$

$$\leq \sigma_{\delta,\tau}(n) \cdot (n-1) \cdot F_\lambda^{n-2}(\tau + \sigma_{\delta,\tau}(n)) \cdot f_\lambda(\tau).$$

Plugging this back in gives that

$$S \leq \sum_{n=1}^{\infty} \sigma_{\delta,\tau}(n) \cdot (1 - F_\lambda(\tau)) \cdot \sigma_{\delta,\tau}(n) \cdot (n-1) \cdot F_\lambda^{n-2}(\tau + \sigma_{\delta,\tau}(n)) \cdot f(\tau)$$

$$= \sum_{n=1}^{\infty} \sigma_{\delta,\tau}(n)^2 \cdot (1 - F_\lambda(\tau)) \cdot (n-1) \cdot F_\lambda^{n-2}(\tau + \sigma_{\delta,\tau}(n)) \cdot f_\lambda(\tau)$$

$$= \sum_{n=1}^{\infty} 256 \cdot \tau^2 r_\delta(n)^2 \cdot e^{-\lambda\tau} \cdot (n-1) \cdot (1 - e^{-\lambda(\tau + \sigma_{\delta,\tau}(n))})^{n-2} \cdot \lambda \cdot e^{-\lambda\tau}$$

Now, observe that $r_\delta(n)^2 \leq C\frac{\log(n/\delta)}{n}$ for some universal constant $C$. Hence,

$$S \leq \frac{256 \cdot C\tau^2\lambda}{e^{2\lambda\tau}} \sum_{n=1}^{\infty} \log(n/\delta) \cdot (1 - e^{-\lambda(\tau + 16\tau \cdot r_\delta(n))})^{n-2}.$$

It suffices to upper bound

$$\sum_{n=1}^{\infty} \log(n/\delta) \cdot (1 - e^{-\lambda(\tau + 16\tau \cdot r_\delta(n))})^{n-2}.$$

Define $A := e^{-\lambda\tau}$ and $q_n := 1 - e^{-\lambda(\tau + 16\tau \cdot r_\delta(n))}$. Then, observe that

$$q_n = 1 - Ae^{-16\lambda\tau \cdot r_\delta(n)}.$$

We want to find the smallest $N$ such that for all $n \geq N$, we get

$$16\lambda\tau \cdot r_\delta(n) \leq \log 2.$$

Substituting the definition of $r_\delta(n)$ and solving for $n$ gives that we need to set

$$N \geq \Theta\left(\lambda^2\tau^2 \log(1/\delta)\right).$$

For this choice of $N$, we have that for all $n \geq N$,

$$e^{-16\lambda\tau \cdot r_\delta(n)} \geq \frac{1}{2} \implies q_n \leq 1 - \frac{A}{2}.$$

Thus, for all $n \geq N$, we have that

$$q_n^{n-2} \leq \left(1 - \frac{A}{2}\right)^{n-2}.$$

Now, split the infinite sum

$$\sum_{n=1}^{\infty} \log(n/\delta) \cdot (1 - e^{-\lambda(\tau + 16\tau \cdot r_\delta(n))})^{n-2} = \sum_{n=1}^{\infty} \log(n/\delta) \cdot q_n^{n-2} = \sum_{n<N} \log(n/\delta) \cdot q_n^{n-2} + \sum_{n>N} \log(n/\delta) \cdot q_n^{n-2}.$$

We can trivially bound

$$\sum_{n<N} \log(n/\delta) \cdot q_n^{n-2} \leq N\log(N/\delta).$$

As for the second sum, we claim that

$$\sum_{n>N} \log(n/\delta) \cdot q_n^{n-2} \leq \sum_{n=1}^{\infty} \log(n/\delta)\left(1 - \frac{A}{2}\right)^{n-2} \leq O\left(\frac{\log(2/\delta A)}{A}\right) = O\left(\lambda\tau \cdot e^{\lambda\tau} \cdot \log(1/\delta)\right)$$

To see why the second inequality is true, let $q := 1 - \frac{A}{2}$. Then,

$$\sum_{n=1}^{\infty} \log(n/\delta)\left(1 - \frac{A}{2}\right)^{n-2} = \frac{1}{q^2} \sum_{n=1}^{\infty} \log(n/\delta)q^n.$$

We will focus on bounding $\sum_n \log(n/\delta)q^n$. Define $p_n = (1-q)q^{n-1}$. Note that $p_n$ is the PMF of a geometric distribution $p$ with parameter $1 - q$. Then,

$$\sum_{n=1}^{\infty} \log(n/\delta)q^n = \frac{q}{1-q} \sum_{n\geq 1} p_n \log(n/\delta) = \frac{q}{1-q} \mathbb{E}_{n\sim p}\left[\log(n/\delta)\right].$$

By Jensen's inequality and the fact that $\log(x/\delta)$ is concave, we have that

$$\mathbb{E}_{n\sim p}\left[\log(n/\delta)\right] \leq \log\left(\mathbb{E}_{n\sim p}\left[n/\delta\right]\right) = \log\left(\frac{1}{\delta(1-q)}\right).$$

Hence, we have that

$$\sum_{n=1}^{\infty} \log(n/\delta)q^n \leq \frac{q}{1-q} \log\left(\frac{1}{\delta(1-q)}\right)$$

and

$$\frac{1}{q^2} \sum_{n=1}^{\infty} \log(n/\delta)q^n \leq \frac{1}{q(1-q)} \log\left(\frac{1}{\delta(1-q)}\right).$$

Since $A \leq 1$, we have that $q \geq \frac{1}{2}$, thus,

$$\frac{1}{q^2} \sum_{n=1}^{\infty} \log(n/\delta)q^n \leq \frac{2}{(1-q)} \log\left(\frac{1}{\delta(1-q)}\right).$$

Plugging in $q = 1 - \frac{A}{2}$ completes the claim that

$$\sum_{n=1}^{\infty} \log(n/\delta) \left(1 - \frac{A}{2}\right)^{n-2} \leq O\left(\frac{\log(2/\delta A)}{A}\right) = O\left(\lambda \tau \cdot e^{\lambda \tau} \cdot \log(1/\delta)\right),$$

where the last equality follows from the definition of $A$. Now, returning to the original proof, we have that

$$\sum_{n=1}^{\infty} \log(n/\delta) \cdot (1 - e^{-\lambda(\tau + 16\tau \cdot r_\delta(n))})^{n-2} \leq \widetilde{O}_{\delta,\tau,\lambda}(\lambda^2 \tau^2) + \widetilde{O}_\delta\left(\lambda \tau \cdot e^{\lambda \tau}\right).$$

Multiplying by the outer factor gives that

$$S \leq \frac{256 \cdot C \tau^2 \lambda}{e^{2\lambda \tau}} \left(\widetilde{O}_{\delta,\tau,\lambda}(\lambda^2 \tau^2) + O\left(\lambda \tau \cdot e^{\lambda \tau}\right)\right)$$

$$= \widetilde{O}_\delta \left(\frac{\lambda^3 \tau^4}{e^{2\lambda \tau}} + \frac{\lambda^2 \tau^3}{e^{\lambda \tau}}\right)$$

$$= \widetilde{O}_\delta \left(\frac{\lambda^2 \tau^3}{e^{\lambda \tau}} \left(1 + \frac{\lambda \tau}{e^{\lambda \tau}}\right)\right)$$

$$= \widetilde{O}_\delta \left(\frac{\lambda^2 \tau^3}{e^{\lambda \tau}}\right)$$

$$= \widetilde{O}_\delta \left(\frac{(\lambda \tau)^3}{\lambda \cdot e^{\lambda \tau}}\right)$$

$$= \widetilde{O}_\delta \left(\frac{1}{\lambda}\right)$$

The third fifth equality follows from the fact that $\frac{x}{e^x} = O(1)$ and $\frac{x^4}{e^x} = O(1)$ respectively. This completes the proof. $\qquad\square$

## F    ADDITIONAL EXPERIMENTAL RESULTS

This appendix provides comprehensive experimental results across all datasets, language models, and reward models. While Section 5 focused on representative examples, here we present the complete evaluation demonstrating the consistency and robustness of our adaptive approach. All empirical results in this section fix $\alpha = 0.99$. We study the impact of other choices for $\alpha$ in Appendix F.5.

### F.1    EXPERIMENT 1: PROFIT PERFORMANCE ACROSS ALL CONFIGURATIONS

We evaluate how consistently our adaptive algorithm matches optimal non-adaptive performance across diverse settings. The profit performance ratio is defined as:

$$\text{Profit Performance Ratio} = \frac{\text{Profit of Adaptive Best-of-N}}{\text{Profit of Best Non-adaptive Best-of-N}}$$

where profit equals utility minus total generation cost. A ratio near 1.0 indicates the adaptive algorithm matches the best possible non-adaptive strategy without requiring hyperparameter tuning.

The results, shown in Figure 5 for the AlpacaFarm and HH-RLHF datasets with RM-Mistral-7B and FsfairX-LLaMA3-RM-v0.1, confirm the adaptive algorithm's strength.

The adaptive algorithm outperforms most of the time even the best-tuned generator dependent non-adaptive strategy. This implies that its dynamic behavior is more profitable than any fixed, one-size-fits-all approach.

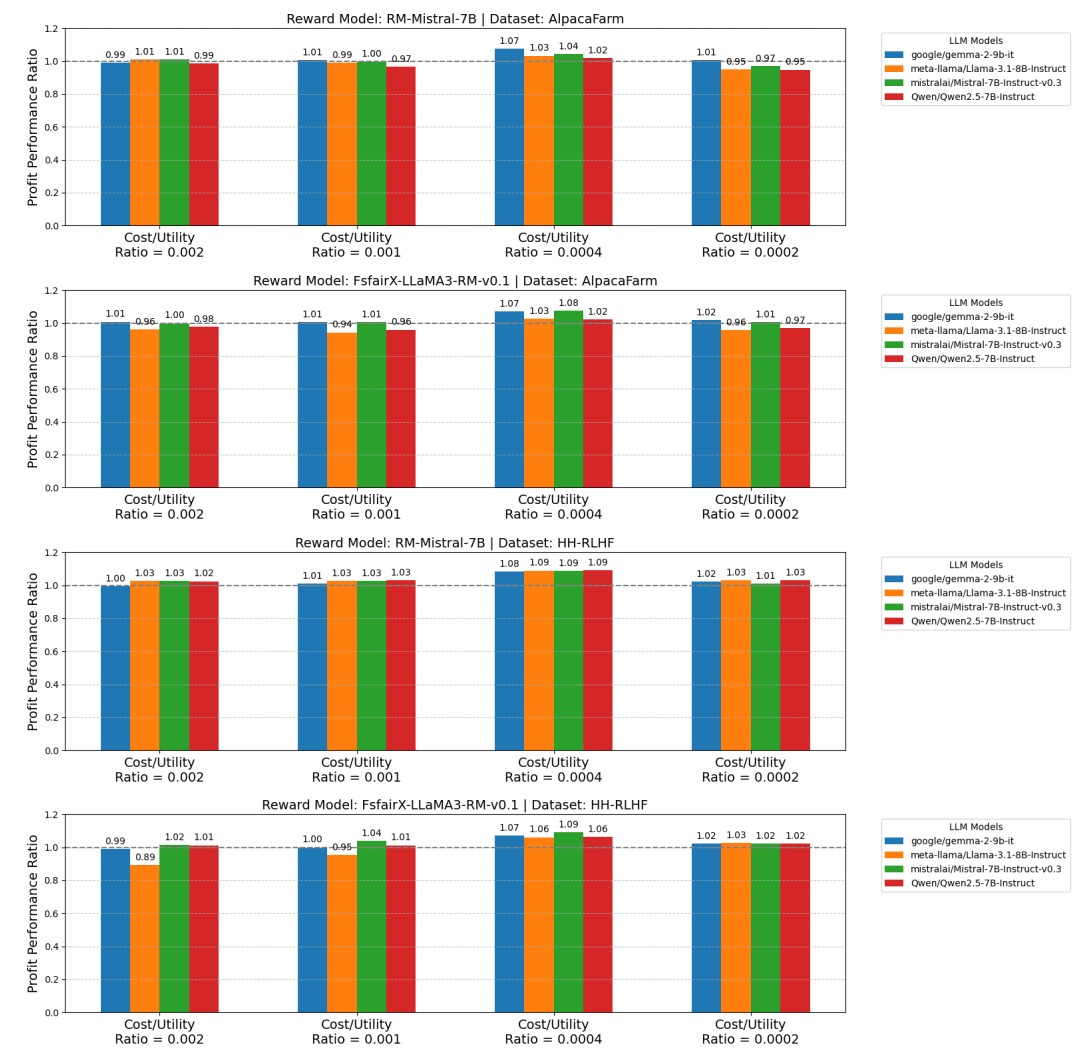

Figure 5: Profit performance ratios for four LLM generators across varying cost/utility ratios. Each panel represents a different dataset-reward model combination, with four cost/utility scenarios per panel. Higher ratios indicate better profit efficiency relative to best non-adaptive algorithm.

## F.2 EXPERIMENT 2: WIN RATE ANALYSIS ACROSS CONFIGURATIONS

We compare head-to-head performance when adaptive and non-adaptive algorithms use identical computation budgets.

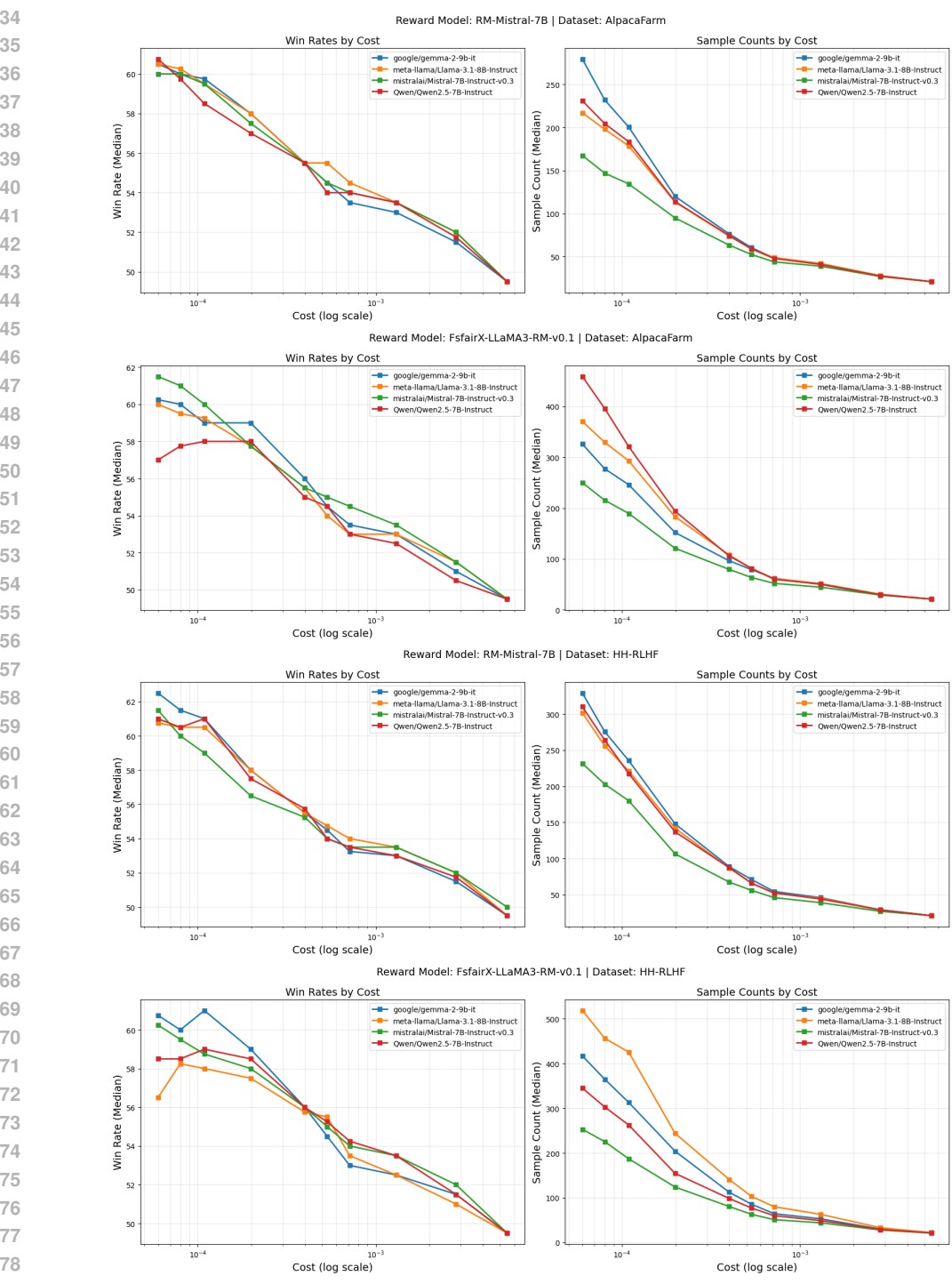

Figure 6: Performance evaluation of adaptive versus non-adaptive generation under matched computational budgets. Each row corresponds to a unique reward model-dataset configuration. Left panels present win rates computed from 100 sample permutations where adaptive algorithms compete against non-adaptive baselines. Right panels show mean sample counts (identical for both strategies due to budget matching) averaged over the same 100 permutations. Both metrics are plotted against cost (log scale, $10^{-5}$ to $10^{-2}$) and represent median values across all evaluation prompts.

Figure 6 reveals consistent patterns across all settings:

- **Budget scaling:** The adaptive algorithm's advantage grows with the available budget. Its win rate increases from around 50% (at chance) with minimal resources to over 54% when the budget exceeds 100 samples.

- **Model consistency:** The win rate trends are remarkably consistent across different models. This demonstrates that the algorithm's effectiveness is not tied to a specific model.

- **Dataset effects:** Similarly, the performance curves hold steady across different datasets, proving the algorithm is robust to variations in input data and prompt styles.

This consistent advantage across diverse models, datasets, and budgets validates our central hypothesis: the adaptive algorithm succeeds by exploiting the unique reward distribution of each prompt, rather than relying on a specific setup.

F.3 EXPERIMENT 3: EFFICIENCY GAINS AT TARGET QUALITY LEVELS

This experiment quantifies the computational efficiency of our adaptive algorithm. We measure how many fewer samples it needs than a non-adaptive approach to reach the same quality, defined by the acceptance rate.

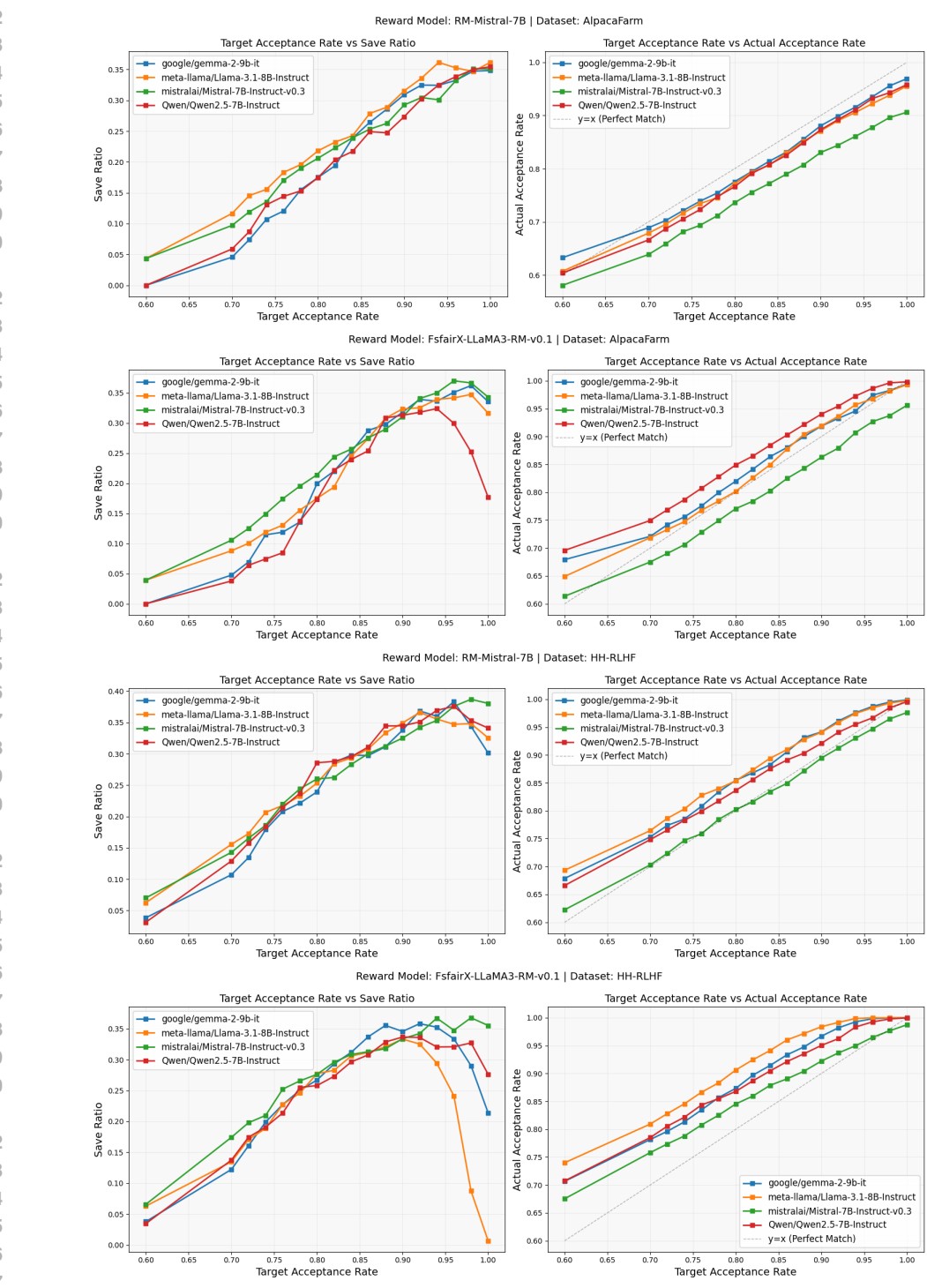

Figure 7: Computational savings from adaptive generation while maintaining acceptance rate equivalence. Each row presents a unique reward model-dataset pairing. Left panels quantify the save ratio—the fraction of samples eliminated by adaptive algorithms compared to non-adaptive methods when both achieve identical average acceptance rates. Specifically, for each target acceptance rate, the adaptive algorithm yields an actual acceptance rate, and the non-adaptive sample count is calibrated to match this actual rate. Right panels show the relationship between target and actual acceptance rates for the adaptive algorithm, illustrating calibration behavior. Save ratios are averaged over 100 generation stream permutations, with all metrics indexed by target acceptance rate (0.6–1.0) and aggregated via median across 100 test prompts.

Figure 7 demonstrates two critical findings:

**Right panels (Target vs Achieved Rates):** The adaptive algorithm effectively follows the desired quality targets. While a small gap exists between the target and achieved acceptance rates, the graphs confirm that our algorithm reliably adjusts its behavior to closely approximate the specified quality level across all configurations.

**Left panels (Compute Savings):** The algorithm delivers significant efficiency gains, with savings that peak when targeting high levels of quality. The trend is as follows:

- Savings increase from 10% at a moderate quality target (0.70 acceptance rate) to a peak of $\sim 30\%$ for near-optimal targets (0.90+ acceptance rate).

- However, savings decrease as the target approaches 100%. This is because the extreme quality requirement forces the adaptive algorithm to use its maximum sample budget, causing its behavior to converge with the non-adaptive baseline. Importantly, even in this scenario, it never performs worse.

This peak in savings at high (but not perfect) quality levels demonstrates the algorithm's core strength: its ability to recognize when a near-optimal response has been found and stop generation early. In contrast, non-adaptive methods must always continue sampling to maintain the same quality guarantee.

Efficiency gains increase monotonically with target quality:

- Roughly 10% savings at 0.70 acceptance rate (moderate quality)

- Roughly 20% savings at 0.80 acceptance rate (good quality)

- Roughly 30% savings at 0.90+ acceptance rate (near-optimal quality)

- When 100% acceptance rate is targeted, our experimental setup suffers from maximum sample size leading that adaptive algorithm becomes closer to non-adaptive one. This explains decrease on save ratio as we get close to 100% target acceptance rate though it never performs worse than non-adaptive algorithm with same computation budget.

The increasing savings at higher quality levels reflect the adaptive algorithm's ability to recognize when it has likely found a near-optimal response, while non-adaptive methods must continue sampling to maintain guarantees.

F.4    CROSS-DATASET AND CROSS-MODEL INSIGHTS

A cross-experimental analysis of our results reveals three consistent and noteworthy findings:

- **Robustness Across Configurations:** The performance metrics and advantages of the adaptive algorithm remained consistent across all 1,600 unique generation profiles. This demonstrates that the approach generalizes effectively and is not over-fit to specific model-dataset pairings.

- **Positive Scaling with Budget:** The superiority of the prompt-adaptive algorithm becomes more pronounced as the number of candidate generations increases, indicating that its advantages scale positively with a larger computational budget.

- **Reward Model Independence:** The performance trends were congruent for both the RM-Mistral-7B and FsfairX-LLaMA3-RM-v0.1 reward models. This suggests that the benefits of the adaptive strategy are independent of the specific reward function employed.

F.5    IMPACT OF $\alpha$

In this section, we present empirical performance of UCB Pandora's Box (Algorithm 1) for gold-standard quantiles $\alpha \in \{0.50, 0.75\}$. From Figures 8 and 9, we find that Algorithm 1 is robust to the choice of $\alpha$ and obtains similar empirical results to when $\alpha = 0.99$. Similar results are observed for the HH-RLHF dataset and FsfairX-LLaMA3 reward model.

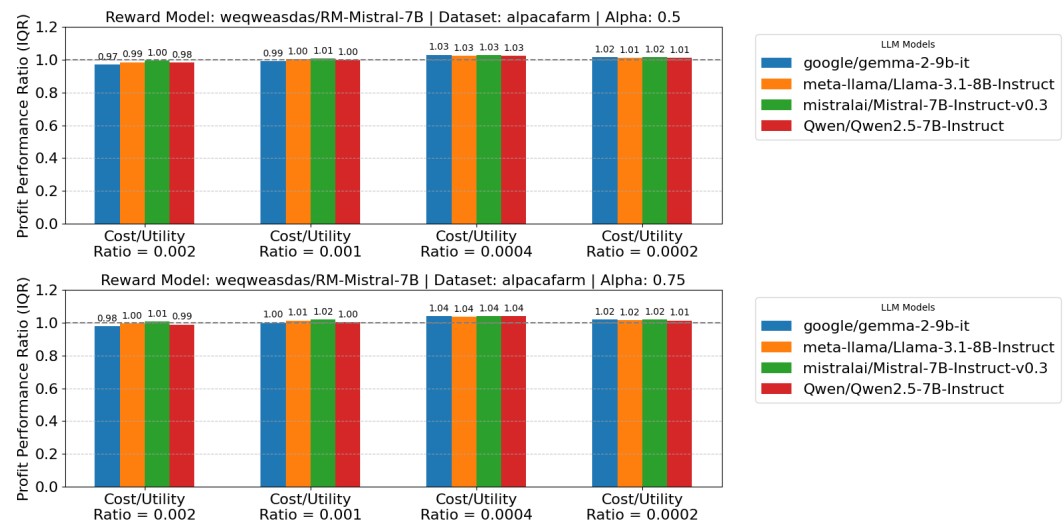

Figure 8: PPRs of UCB Pandora's Box (Algorithm 1) for gold-standard quantiles $\alpha \in \{0.50, 0.75\}$ on the AlpacaFarm dataset and RM-Mistral-7B reward model.

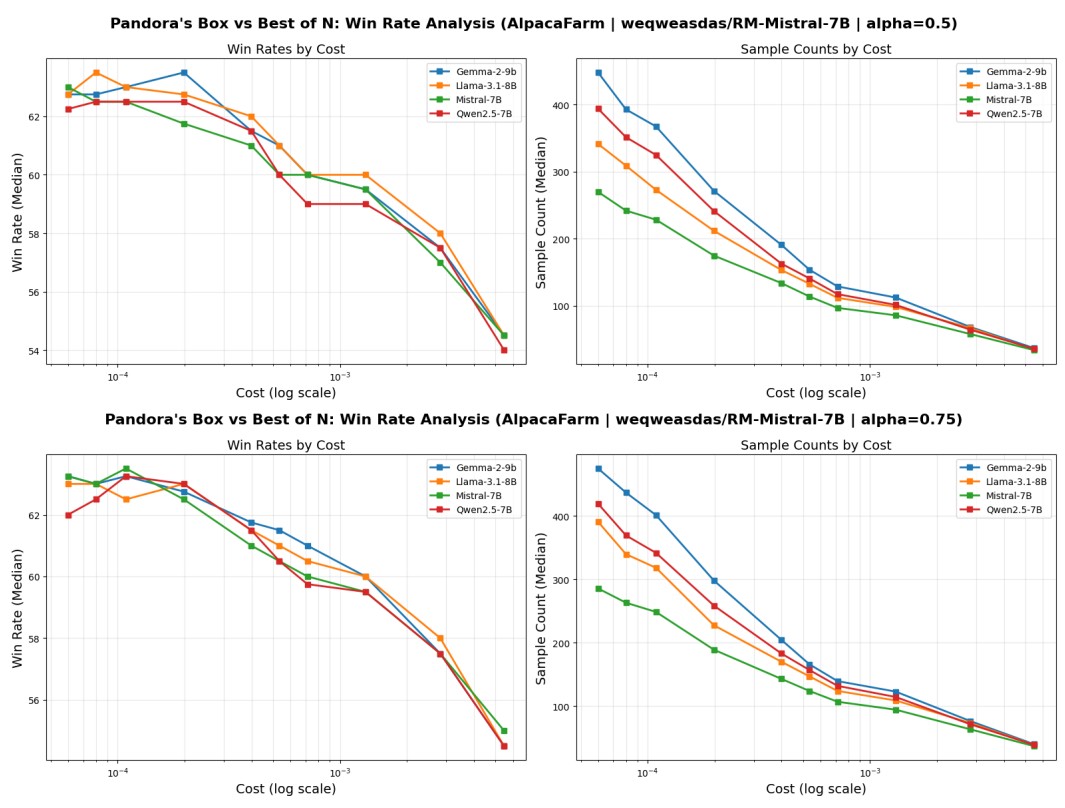

Figure 9: Win rates and sample counts of UCB Pandora's Box (Algorithm 1) for $\alpha \in \{0.50, 0.75\}$ on the AlpacaFarm dataset and RM-Mistral-7B reward model.

## F.6 QQ-PLOTS

In this section, we empirically corroborate our choice to use the Exponential distribution to fit the right-tail of the exponentiated rewards. Figure 10 presents QQ-plots of the exponentiated rewards

against a (shifted) exponential distribution on four prompts from the AlpacaFarm dataset for the RM-Mistral-7B reward model. The almost linear nature of the blue dots indicate that the Exponential distribution generally fits the right-tail of the exponentiated rewards well. Similar results are observed for other prompts in the AlpacaFarm and HH-RLHF datasets as well as the FsfairX-LLaMA3 reward model.

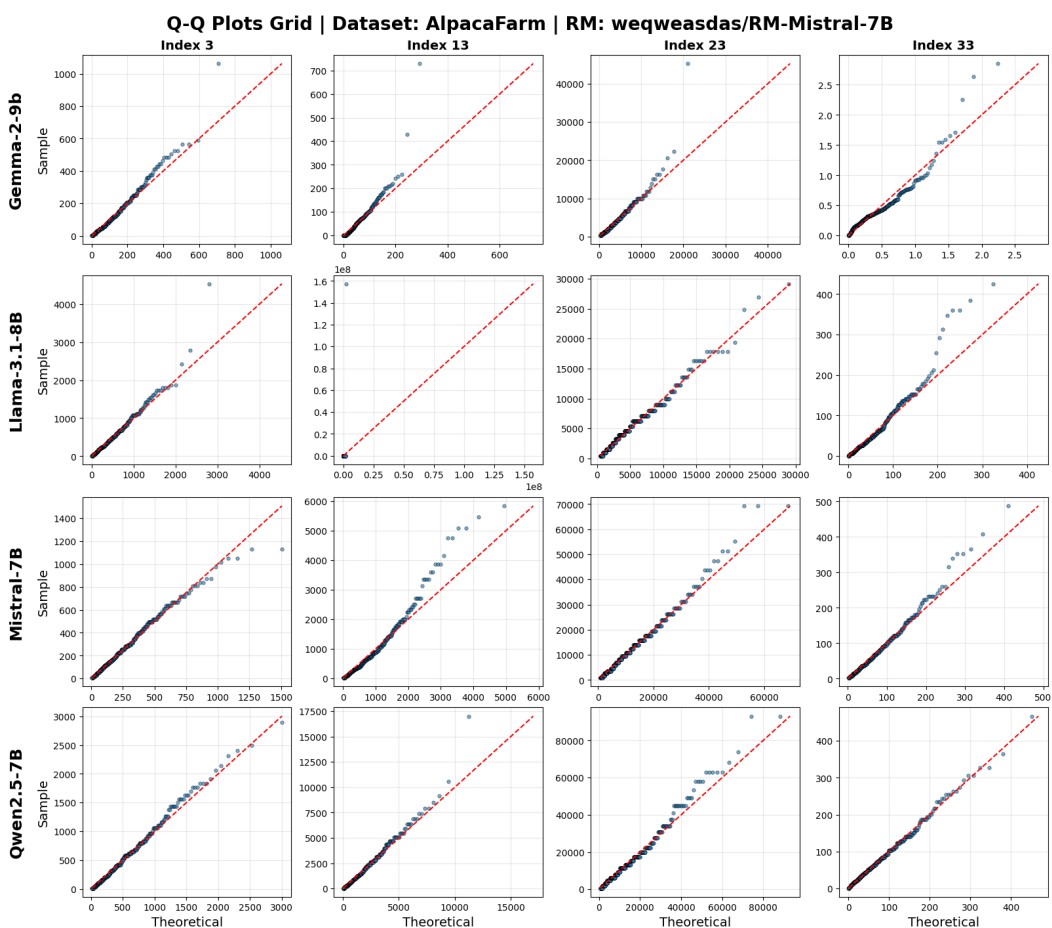

Figure 10: QQ-plots of the right-tail of the exponentiated rewards from the RM-Mistral-7B reward model against a shifted Exponential distribution for four prompts from the AlpacaFarm dataset.

### F.7 COMPARISON WITH NAIVE BASELINE

In this section, we compare our method against a naive adaptive baseline which stops sampling once the maximum reward seen so far has not changed for 5 steps. Figure 11 presents the median profit of Algorithm 1, the naive adaptive baseline, and the profit of the non-adaptive BoN (for various $N$ values) across prompts from the AlpacaFarm dataset using the the RM-Mistral-7B reward model. We find that across a wide range of cost values and LLMs, the naive baseline obtains smaller profit than our adaptive method. Moreover, this difference in profit between Algorithm 1 and the naive baseline increases as the cost/utility ratio decreases. Similar results are observed for the HH-RLHF dataset and FsfairX-LLaMA3 reward model.

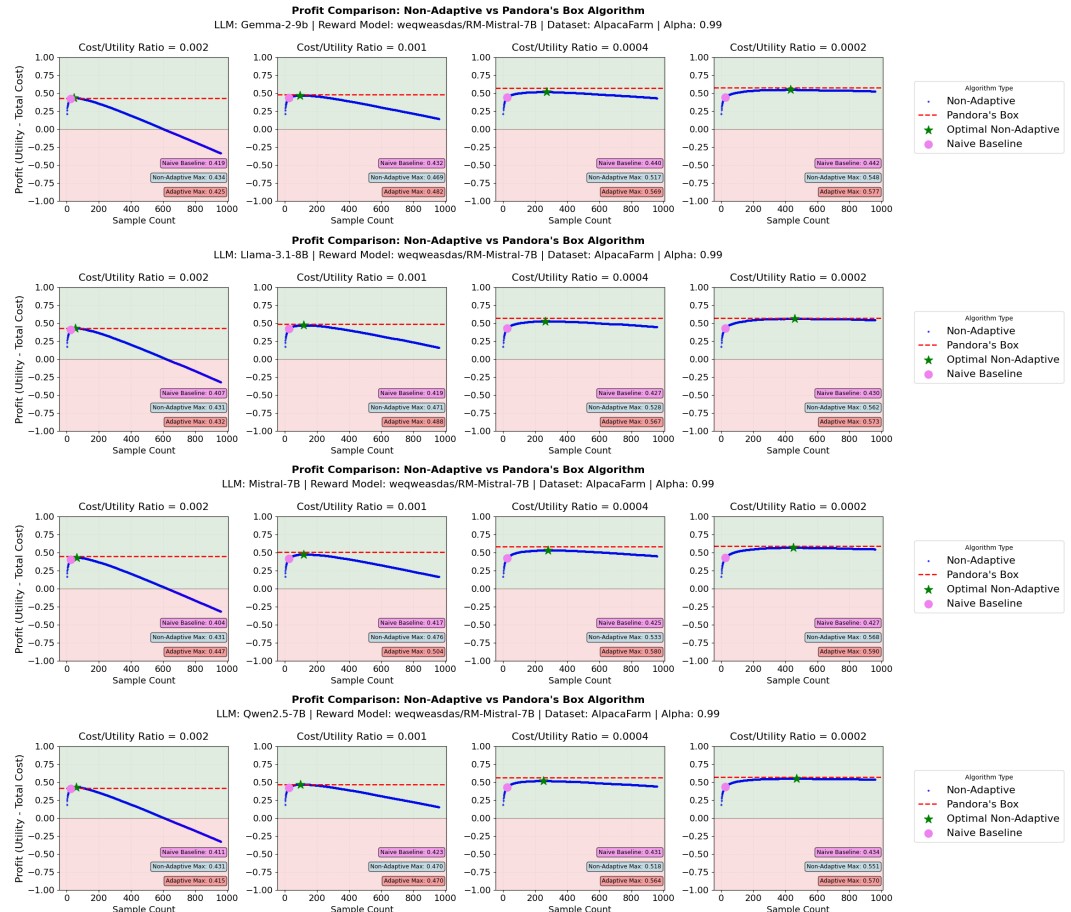

Figure 11: Median profit (over prompts from the AlpacaFarm dataset) of UCB Pandora's Box (Algorithm 1), the naive baseline, and the non-adaptive BoN (at various $N$ values) for cost/utility ratios $c \in \{0.002, 0.001, 0.0004, 0.0002\}$.

