# OpenReview forum: "Optimal Stopping vs Best-Of-$N$ for Inference Time Optimization"
_ICLR.cc/2026/Conference — Submitted to ICLR 2026_

### Official Review · Reviewer_pf2g · 2025-10-25

**Soundness:** 2
**Presentation:** 3
**Contribution:** 2
**Rating:** 4
**Confidence:** 4

**Summary:**

The paper reframes Best-of-N as an optimal-stopping problem. The goal is to learn a fair-cap threshold online for a given prompt and stop once the running best reward crosses it. They estimate this threshold with a UCB-style rule and use a Bradley–Terry–style transform centered at a per-prompt quantile to normalize rewards across prompts. This adaptive policy is shown to achieve the same performance as a non-adaptive Best-of-N policy while requiring 15-35% fewer generations on average.

**Strengths:**

- Casting BoN as a fair-cap stopping problem is intuitive and principled. It introduces a formal decision-theoretic framework for inference-time alignment by providing  a concrete “when to stop” rule instead of grid-searching for N.
- I appreciate the thoughtfulness in considering practical challenges like cross-prompt reward miscalibration, as well as setting an acceptance-rate target.
- The empirical experiments are extensive, illustrative, and useful.

**Weaknesses:**

- My main concern is that best of $N$ is not run sequentially in practice. Modern inference systems are heavily optimized for parallel, batched decoding to maximize hardware utilization. An adaptive, sequential approach trades total compute for potentially higher latency and lower system throughput. This is why BoN is very attractive as an inference-time method, aside from its simplicity and effectiveness. The optimal stopping method adds more complexity because of the uncertainty in $N$, potentially making inference much less efficient.

- The sample counts (the $N$ in BoN) are large and unrealistic (> 50 in Figures 1 and 2). This may suggest (if I understand correctly) that the cost the authors fix to show the efficiency of their algoirthm may be unrealistic.

- There is a significant gap between the theory presented and the practical version of the algorithm. I appreciate the author's discussion of their algorithm that uses an explicit cost c and max utility B, which are acknowledged to be difficult to specify. However, what is actually deployable in practice is the target-acceptance variant. I believe this aspect was not discussed sufficiently, as it underpins the performance of their algorithm. This disconnect weakens the claim that the practical algorithm is principled and provably near-optimal.

- The practical algorithm's performance hinges on a strong, unmotivated parametric assumption: that the right tail of the (exponentiated) reward distribution can be modeled by a shifted exponential. The paper does not provide an empirical validation (e.g., goodness-of-fit tests) for this assumption. If there is a model mismatch, the UCB estimates for the fair-cap value could be invalid.

- The paper's novelty claims should be framed more precisely. I do not believe that the first contribution (claiming that you present the "the first stopping strategy that adapts to unknown reward distributions") is factual. The novelty is better framed as first application to LLM inference-time alignment, not the learning-theoretic idea itself.

**Questions:**

Q1: Qualitatively or quantitatively, can you discuss the tradeoffs between parallel BoN and sequential inference?

Q2: using the data collected for Figure 4, can you show how well the shifted exponential fit is on average per prompt?

Q3: You set κ to the prompt-wise α-percentile and then map via AR_κ(v). This ties the utility to a quantile that itself is estimated online. This is confusing to me. This seems to say that the utility function is itself non-stationary. How does this reconcile with the theoretical analysis in Section 3, which assumes a fixed, albeit unknown, reward distribution?

Suggestion 1: Aside from fixed N, you could consider a baseline that stops after the reward has not improved by a meaningful amount for k consecutive steps, or one that stops after crossing a fixed reward threshold. Including such baselines would better isolate the specific benefits of the optimal stopping framework, which I do not exactly see. I would also test the impact of uncalibrated rewards to convince the readers of the usefulness of your BT-normalization compared to other standard techniques.

Suggestion 2: The current experiments evaluate the algorithm's end-to-end performance, but it is difficult to assess the accuracy of the underlying estimation of the fair-cap value τ, as the ground truth is unknown. Have you considered a setup with a verifiable reward (such as math)? I believe the usefulness of optimal stopping might be more apparent there.

Suggestion 3: I believe the paper is missing a few important references. One is AdaBoN [1] which is similar in spirit to your paper (posted on arxiv on May 2025). They also estimate the reward distribution online and then adaptively allocate the budget. Also, [2] train a model that predicts the expected marginal gain in reward from allocating additional responses to a prompt. I appreciate if you would discuss how your work compares to them. Moreover, [3] give a way to find the an optimal $N$ offline for BoN (posted on arxiv on June 2025)..However, they require access to a true reward calibration dataset.

[1] Raman, V., Asi, H., & Kale, S. (2025). AdaBoN: Adaptive Best-of-N Alignment. arXiv preprint arXiv:2505.12050.

[2] Damani, M., Shenfeld, I., Peng, A., Bobu, A., & Andreas, J. (2024). Learning how hard to think: Input-adaptive allocation of lm computation. COLM 2025.

[3] Khalaf, H., Verdun, C. M., Oesterling, A., Lakkaraju, H., & du Pin Calmon, F. (2025). Inference-Time Reward Hacking in Large Language Models. NeurIPS 2025.

---

> ### Author Response · Authors · 2025-11-21
>
> Thanks for the review! We've attempted to address the most salient comments below. Please let us know if further questions arise.
>
> >  My main concern is that best of N is not run sequentially in practice.
>
> We refer the reviewer to our general comment above regarding the batched variant of our method. In summary, our framework can be easily generalized to exploit parallel generation by defining a "sample" from the reward distribution to be the maximum reward over a batch of rewards.
>
> > The sample counts (the  in BoN) are large and unrealistic (> 50 in Figures 1 and 2). This may suggest (if I understand correctly) that the cost the authors fix to show the efficiency of their algorithm may be unrealistic.
>
> This is a great question. Yes, for very large models like ChatGPT and Gemini, performing $> 50 $ forward passes may not reasonable. However, there is a growing interest in on-device/edge computation, where smaller models are necessary. Here, it is reasonable to be able generate a large number of responses in order to stay competitive with RL-based post-training methods.  In addition, for complex reasoning tasks (e.g., mathematics, coding) one often prioritizes final accuracy over immediate latency. In these scenarios—where compute is explicitly traded for correctness—our method provides a principled guardrail to ensure resources are not wasted on problems that are solved early.
>
> > There is a significant gap between the theory presented and the practical version of the algorithm. I appreciate the author's discussion of their algorithm that uses an explicit cost c and max utility B, which are acknowledged to be difficult to specify. However, what is actually deployable in practice is the target-acceptance variant. I believe this aspect was not discussed sufficiently, as it underpins the performance of their algorithm. This disconnect weakens the claim that the practical algorithm is principled and provably near-optimal.
>
> We thank the reviewer for this comment, and for appreciating the target-acceptance variant. We will make sure to expand more about this variant in the final version. That said, we believe that the cost variant is still meaningful and easy to implement. In particular, it is easy to set $c$ in practice -- it is the fraction of the payoff of an accepted response that it costs to do one forward pass through both the LLM and reward model. The actual cost $c'$ of a forward pass through the LLM and reward model is normally known. Hence, once one estimates the value $B$ of a user accepting a response,  $c$ is just  $c'/B$.
>
> > The practical algorithm's performance hinges on a strong, unmotivated parametric assumption: that the right tail of the (exponentiated) reward distribution can be modeled by a shifted exponential. The paper does not provide an empirical validation (e.g., goodness-of-fit tests) for this assumption. If there is a model mismatch, the UCB estimates for the fair-cap value could be invalid.
>
> We refer the reviewer to our general comment above regarding QQ plots. In summary, we have uploaded a revised manuscript and have included them in Appendix F.6.
>
> > The paper's novelty claims should be framed more precisely. I do not believe that the first contribution (claiming that you present the "the first stopping strategy that adapts to unknown reward distributions") is factual. The novelty is better framed as first application to LLM inference-time alignment, not the learning-theoretic idea itself.
>
>  We thank the reviewer for this comment. We agree that this line as written is incorrect. In the final manuscript, we will revise this line so that it reads ``We propose the first stopping strategy for the Pandora's Box problem with identical but unknown reward distributions." Surprisingly, to the best of our knowledge, we were unable to find prior work that studies this basic optimal stopping problem in the unknown distribution case. So, in this regard, our theoretical results are novel and of independent interest.

---

> > ### Author Response · Authors · 2025-11-21
> >
> > > You set k to the prompt-wise alpha-percentile and then map via ARk(v). This ties the utility to a quantile that itself is estimated online. This is confusing to me. This seems to say that the utility function is itself non-stationary. How does this reconcile with the theoretical analysis in Section 3, which assumes a fixed, albeit unknown, reward distribution?
> >
> > The true $\alpha$-percentile of the reward distribution is not random, but a fixed, unknown quantity. Hence, the distribution of  $\alpha$-transformed rewards is fixed, but unknown to us. In practice, we do not know the  $\alpha$-percentile of the reward distribution, and this must also be estimated from the data. Hence, we are not able to exactly compute the  $\alpha$-transformed reward (even though it exists!), but can estimate it. Note that any meaningful reward transformation that normalizes rewards consistently across different prompts will need to transform the reward using unknown quantities that depend on the reward distribution itself. Hence, the inability to compute the actual transformed reward is inevitable.
> >
> > >  Suggestion 1: Aside from fixed N, you could consider a baseline that stops after the reward has not improved by a meaningful amount for k consecutive steps, or one that stops after crossing a fixed reward threshold. Including such baselines would better isolate the specific benefits of the optimal stopping framework, which I do not exactly see. I would also test the impact of uncalibrated rewards to convince the readers of the usefulness of your BT-normalization compared to other standard techniques.
> >
> > This is a great suggestion. In our revised manuscript, we have included a baseline which stops if the maximum rewards has not changed in $k$ steps. These results are in Appendix F.7 of the revised manuscript. We find that this baseline consistently performs worse than our adaptive method, and especially so at small cost values.
> >
> > That said, Figures 1 and 5 show that our adaptive method is as profitable as the optimal non-adaptive Best-of-$N$ strategy in hindsight. This already highlights the importance of adaptivity when aiming to maximize profit since the optimal $N$ to use for a prompt is not known apriori. Regarding the usefulness of normalization, note that some reward models output negative values, as their outputs are only meaningful comparatively. Since $c > 0$, the fair-cap value of the raw reward distribution may not be well-defined, resulting in a degenerate policy that stops immediately and never samples. On the other hand, our BT-normalization results in a transformed reward distribution whose support is always between $(0, 1).$ Moreover, the scale of these transformed reward distributions are consistent across different prompts, allowing the cost $c$ to be chosen meaningfully. This simple example highlights the usefulness and necessity of reward normalization.
> >
> > > Suggestion 2: The current experiments evaluate the algorithm's end-to-end performance, but it is difficult to assess the accuracy of the underlying estimation of the fair-cap value tau, as the ground truth is unknown. Have you considered a setup with a verifiable reward (such as math)? I believe the usefulness of optimal stopping might be more apparent there.
> >
> > We thank the reviewer for this suggestion. We have considered a setup with a verifiable reward, and this is a core focus of our follow-up work. Unfortunately, we will not be able to complete these experiments due to the short rebuttal period.
> >
> > > Suggestion 3: I believe the paper is missing a few important references...
> >
> > We thank the authors for pointing these out. We will make sure to include and discuss them in the final version. Both [1] and [2] are similar in spirit but consider a fundamentally different problem: in these works, there exist a *batch* of prompts and the goal is figure out how to allocate an inference budget across this batch. In contrast, our method operates on a single prompt. As the reviewer noted themselves, the method proposed by [3] requires an external *true* reward calibration dataset (as opposed to proxy rewards from a reward model) and, to the best of our knowledge, is not prompt-adaptive in the sense that the parameters learned from the reward dataset are fixed and used for all future prompts. At a higher-level, [3] is  mainly concerned about reward hacking (i.e. making sure that by maximizing the proxy reward we are truly maximizing the true reward) and not efficiency, while our work is mainly concerned about efficiency and not reward hacking. One interesting direction of future study is that by being prompt-adaptive and stopping early, our methods may naturally also mitigate reward hacking.

---

> > > ### Comment · Reviewer_pf2g · 2025-11-27
> > >
> > > Thanks for your responses! I appreciate your efforts.
> > >
> > > I will maintain my score since I do not find the method generally useful. The increase in performance compared to the naive baseline comes at a much higher inference cost (with $n$ being around 300). I agree that it is reasonable to generate this many model calls in some applications. I would be very excited if in future work you specialize this method directly to small LMs and test different methods to improve performance at inference.

---

> > > > ### Author Response · Authors · 2025-11-28
> > > >
> > > > Thanks for the response! We would like to point out that even at small sample sizes around 100 our method improves on the baseline; see Figure 11 in the Appendix. Notwithstanding, we view our primary contribution to be the framework, which casts adaptive LLM sampling as an optimal stopping problem.

---

### Official Review · Reviewer_N324 · 2025-10-31

**Soundness:** 3
**Presentation:** 4
**Contribution:** 4
**Rating:** 6
**Confidence:** 2

**Summary:**

In this work, the authors model inference time sampling with early stopping as a "Pandora's Box Algorithm", where the goal is to develop an algorithm that decides when to stop generating inference samples and select the best one out of the existing set. The goal of this approach is to get efficiency speedups over having a fixed number of generations N a priori which might be wasteful if the reward-maximizing generation occurs early.

They propose a UCB-Style Pandoras Box algorithm to handle settings where the underlying reward distribution is unknown and use this to create an on-the-fly inference time algorithm that determines when to stop adaptively.

**Strengths:**

* The framing of the problem is novel and highlights an area of improvement for the design of inference time algorithms
* The authors provide theoretical insights into the optimality gap of their approach.
* Through experiments, the method is demonstrated to have significant savings in terms of number of samples compared to Best-of-N.

**Weaknesses:**

* I'm not convinced that the Bradley-Terry Transformation is an important contribution beyond what allows you to prove about your algorithm. Prior works have used the Probability Integral Transform to normalize rewards, both theoretically and empirically, with success [1]. However, this is framed (in the abstract especially) as a core contribution of this work. I would like to see more commentary on the importance of this framework for future research.
* Additionally, it may be worth addressing [1] because they propose a randomized stopping method, which seems (slightly) relevant to your discussion on the limitations of stopping at a fixed N.


Small Notation Comments:
* Line 128: "with N fixed in advance. Moreover, in practice, N is typically fixed" is repetitive
* In the Algorithm, line 8, should the $\mu$s be $\hat{\mu}$s?

[1] Khalaf, H., Verdun, C. M., Oesterling, A., Lakkaraju, H., & Calmon, F. D. P. (2025). Inference-Time Reward Hacking in Large Language Models. NeurIPS 2025

**Questions:**

* Can you elaborate on 1) if the Bradley-Terry Transformation is uniquely necessary to your results (and conversely, whether the PIT could be used to get the same insights) and 2) the tradeoffs between the two? One benefit I see of this transformation is that you don't need to estimate a CDF.
* Can you also clarify the choice of $\mathcal{F}$? What happens if the true distribution of rewards is vastly different than $\mathcal{F}$? Is there some sort of approximation error or tradeoff to choosing a specific (e.g., Exponential) distribution? All of your results begin with assuming a distributional family and then demonstrate that the anytime UCB policy has a low optimality gap on Weitzman's optimal policy. Can you say anything about using the wrong choice of distributional family in your Algorithm?

---

> ### Author Response · Authors · 2025-11-21
>
> Thanks for the review! We've attempted to address the most salient comments below. Please let us know if further questions arise.
>
> > I'm not convinced that the Bradley-Terry Transformation is an important contribution beyond what allows you to prove about your algorithm.
>
> We agree with the reviewer and will de-emphasize this as a contribution in the final version.
>
> > Additionally, it may be worth addressing [1] because they propose a randomized stopping method, which seems (slightly) relevant to your discussion on the limitations of stopping at a fixed N.
>
> We thank the reviewer for pointing out this reference. We were unaware of this at the time of our submission and will make sure to comment on this in the final version. While relevant, the randomized stopping method proposed by [1] is not prompt adaptive -- it simply samples $N'$ from a Poisson distribution with parameter $\mu$, and runs best of $N = N'$. Moreover, to the best of our knowledge, the authors do not discuss how to pick the parameters of the Poisson distribution as a function of the prompt at hand. Instead, the authors require an external reward calibration dataset, consisting of \emph{true} rewards (i.e. not the output of reward models) across different prompts to tune hyperparameters.  In contrast, our method only requires proxy rewards (i.e. rewards from a reward model). At a higher level, their paper is more concerned with reward hacking and does not explicitly consider the cost of generation and reward computation. In contrast, our proposed method focuses mainly on the tradeoff between inference-time cost and maximizing (proxy) reward.
>
> > Can you elaborate on 1) if the Bradley-Terry Transformation is uniquely necessary to your results (and conversely, whether the PIT could be used to get the same insights) and 2) the tradeoffs between the two? One benefit I see of this transformation is that you don't need to estimate a CDF.
>
> This a great point. The reviewer is right that the BT transformation is not uniquely necessary for our results and that any transformation of the rewards that enables a meaningful comparison to the cost can be used. In this way, our main contribution is the generic framework that suggests a principled way to decide when to stop sampling based on optimal stopping theory -- the details of the exact implementation will vary from use case to use case. The PIT transformation suggested by the reviewer is an excellent suggestion and one that we have also thought about. For either transformation, you need to model the reward distribution, since using the empirical CDF (and DKW bounds) would result in a degenerate policy that would either stop immediately, or after a fixed number of rounds. Our choice to do a BT-transformation instead of something like the PIT is due to the fact that it is popular to train reward models on preference data assuming a BT model. For example, both the reward models we consider in this paper were trained using the BT model.
>
> Beyond alignment of reward model training and our transformation, a practical advantage of the BT transformation is its superior sensitivity in the right tail of the distribution. While the PIT tends to saturate for high-quality responses—compressing distinct, excellent candidates into the same high percentile—the BT model preserves their qualitative differences. To illustrate, consider a heavy-tailed reward difference of $b-a=2$; the PIT might treat both samples as indistinguishably high quantiles (e.g., $>0.9$), whereas the BT transformation captures the significant $\approx 88\%$ implied win probability of $b$ over $a$. That said, we are currently finalizing an ablation study using the PIT.
>
> > Can you also clarify the choice of $F$ ? What happens if the true distribution of rewards is vastly different than $F$?
>
> This is a great question. First, as we note in Lines 171-175, if one places no assumptions on $F$, then getting a meaningful bound on the minimax suboptimality gap is hopeless. So, at least theoretically, one must place some distributional assumptions. Nevertheless, the reviewer is correct that in practice, our choice of $F$ may not cover the true underlying reward distribution. Our current theoretical results assume a "realizable setting," where the true distribution selected by the adversary must lie in $F$. Here, it makes sense to bound the sub-optimality gap with respect to Weitzman's algorithm which knows the true $D \in F.$  In the "agnostic" setting, where the adversaries choice of distribution need not lie in $F$, the natural objective is to bound the sub optimality gap with respect to running Weitzman's algorithm on the "best" distribution $D \in F$. This  would be the analogous objective to the sort of generalization/regret bounds one proves in agnostic PAC/online learning. This is a very promising direction of future work, and one that we are actively working on.

---

> > ### Comment · Reviewer_N324 · 2025-11-21
> > **Reviewer Response**
> >
> > Thank you for these responses, they have answered my questions. I understand why the approach I suggested above isn't exactly related and why it is hard to quantify approximation errors in choice of $F$. I find this framing and contribution interesting, and I think as long as you focus on correctly calibrating the framing of your contributions (e.g. the reward normalization approach) this paper merits acceptance to ICLR. It appears I am an outlier in terms of score, so I will keep my score and follow the other discussion closely.

---

### Official Review · Reviewer_7q3p · 2025-11-01

**Soundness:** 3
**Presentation:** 2
**Contribution:** 2
**Rating:** 4
**Confidence:** 3

**Summary:**

The authors introduce an inference‑time optimization framework grounded in the classical Pandora’s Box optimal‑stopping problem, treating each generation as opening a costly “box” with a random reward, and show that an adaptive stopping strategy can match the performance of non‑adaptive Best‑of‑N sampling while using roughly 15–35% fewer generations on average.

**Strengths:**

The paper is original in recasting prompt‑level sampling as a Pandora’s Box optimal‑stopping problem, importing fair‑cap thresholds and Weitzman‑style reasoning to yield an adaptive stopping rule that subsumes fixed-N heuristics and turns test‑time compute into a principled decision problem. It then formalizes the fair‑cap objective, an anytime‑valid UCB construction for the unknown threshold, a general additive‑gap template (Theorem 5) with an explicit exponential‑tail instantiation (Theorem 6, Corollary 7), and a practical algorithm with closed‑form components and small overhead (Algorithm 1). The empirical significance is demonstrated empirically across ~1,600 profiles (100 prompts × 2 datasets × 4 LLMs × 2 reward models): the adaptive policy matches non‑adaptive Best‑of‑(N) while saving roughly 15–35% of generations on average and winning under matched budgets, indicating a relevant method for practitioners seeking an inference‑time alternative to best-of-N.

**Weaknesses:**

**Corollary 7 and "vanishing regret" need reconciliation**

Corollary 7 upper-bounds the sub-optimality by $O_\delta(1/\lambda)$, i.e., a distribution-dependent constant, not a term that vanishes with samples $n$ or confidence $\delta$. Please clarify whether any "vanishing" behavior is intended (e.g., in a different asymptotic or under additional assumptions), and align the abstract/claims ("guarantees vanishing regret relative to Weitzman's optimal policy") with the explicit bound you prove.

**Heavy reliance on modeling assumptions; robustness is unclear**

Guarantees assume (i) i.i.d. draws per prompt, (ii) known per-sample cost $c$, and (iii) a well-specified distribution family $\mathcal{F}$; the worked-out theory and the practical estimator both hinge on a (shifted) exponential tail fit. Could you provide any robustness results (or diagnostics) under tail mis-specification, e.g., when the right tail is heavier/lighter than exponential or multi-modal? Even a lemma or experiment quantifying degradation under controlled mis-fit would help.

**Discussion/conclusion are thin**

I understand the size limitation but the discussion section is not great. The paper doesn't have a proper conclusion and the discussion section reads as brief future-work notes; the paper lacks a proper Conclusion that synthesizes contributions, caveats, and takeaways for practitioners. Adding a short, explicit conclusion would improve readability.

**Related-work coverage is incomplete (see list below)**

Important recent work on inference-time alignment and adaptive compute is missing. Please add and discuss the listed papers; position your method theoretically and empirically relative to them.

- **Adaptive inference-time compute strategies:** (e.g., arXiv:2410.02725, arXiv:2503.01422, arXiv:2412.15287). Please compare assumptions, decision rules, and metrics. If possible, add a small head-to-head on a shared setup.

- **Inference-time alignment variants of Best-of-N:** soft-Best-of-N, Best-of-Poisson (e.g., arXiv:2506.19248, arXiv:2505.03156, arXiv:2507.05913). Ask: Where does your adaptive stopping dominate these stochastic Best-of-N variants at equal compute? Could your UCB fair-cap be combined with soft-aggregation?

**[L127]** "with N fixed in advance. Moreover, in practice, N is typically fixed."

Please rephrase to avoid repetition.

**Questions:**

**[L106]** "We note, however, that Weitzman's algorithm applies to multiple box types."

Could you please spell out the multiple-box setting (heterogeneous boxes) explicitly: assume different reward distributions ($D_1,\dots,D_k$) and opening costs ($c_1,\dots,c_k$)? Are draws independent across boxes?

**[L153]** "Weitzman's celebrated algorithm provides the optimal stopping strategy using fair-cap values when distributions are known."

Please add 2-3 sentences summarizing Weitzman's policy and its assumptions (known $D_i$ and $c_i$, independence, objective). Include a citation to Weitzman (1978) already in your references, plus a general optimal-stopping reference.

**[L173]** "no hope for designing a single, minimax optimal stopping policy S whose additive sub-optimality gap is uniformly bounded across all distributions."

Please define minimax precisely: is this $\inf_S \sup_{D} \mathbb{E}_D[R_W-R_S]$? If so, say so, and clarify that the impossibility is distribution-free over all $D$, motivating restriction to a family $\mathcal{F}$.

**[L174]** "we will assume that we have a known distribution family $\mathcal{F}$ such that the unknown $D\in\mathcal{F}$."

Please list the concrete assumptions on $\mathcal{F}$: identifiability, i.i.d. draws, existence/monotonicity of the fair-cap map $\tau(D,c)$, and that $\mathcal{F}$ admits an anytime-valid confidence sequence for $\tau$ (Def. 4). If you assume parametric (e.g., exponential tail), say so here, not only in §3.2/Alg. 1.

**[L198]** "the confidence parameter is a hyperparameter that influences the exploration-exploitation balance."

Could you please be explicit here? Something like: smaller $\delta$ ⇒ wider UCB on $\tau$ ⇒ more samples (conservative stopping). Is that what you meant? A short sentence quantifying how $\sigma_{\delta,\tau}(n)$ scales with $\delta$ (via $r_\delta(n)$ in Thm. 6) would also be nice.

**[L200–205]** "the fair-cap value often admits a simple monotonic dependence on the distribution's parameters…constructing a confidence sequence for $\tau$ can be reduced to…(1) CB for parameters…(2) propagate through the monotonic mapping."

Please clarify which parameters and the direction of monotonicity. Does monotonicity hold for every parameter or only for a specific scalar (e.g., mean/rate in the exponential)? How about the multi-parametric case?

**[L205 and L243]** The "confidence bounds for parameters → monotone map → confidence bounds for $\tau$" idea appears twice (around L205 and again near the statement/proof of Thm. 5).

Please keep it once and forward-reference the other to reduce redundancy.

**[L229]** You use "family" and later "Exponential distribution family."

Add a remark that *"exponential" here means the Exponential distribution (with rate/scale), not the general exponential family of distributions*. A footnote with a link to a standard reference should be fine.

**[L280]** "practitioners may adjust this parameter based on quality requirements."

What is your take on that? What is your opinion on how this is usually done? Please offer guidance.

**[L300]** "This transformation maps rewards into $[0,1]$."

Can $\mathrm{AR}_\kappa$ ever be exactly 0? Only in the limit $v\to-\infty$; for finite $v$ it's $>0$. Also, please explain the factor 2: it makes $\mathrm{AR}=1$ since the Bradley–Terry term equals 1/2 at $v=\kappa$.

**[L302]** "acceptance rate approximates the probability that an end-user accepts the response"

Confirm this refers to the particular sample with reward $v$. A small rephrase like "a response with reward $v$" would remove ambiguity.

**[L324]** "we exponentiate the rewards and fit a shifted exponential to the right-tail (above median)."

How many tail samples do you require before fitting? Please specify the minimum $t$ (Alg. 1), report sensitivity to $t$, and add a sanity check like a QQ-plot. Could you specify a rule-of-thumb to help practitioners?

**[L362–363]** "Streaming updates eliminate redundant computation"

Could be clarified with one sentence on what you cache.

**[L364]** "$\alpha$-percentile … computed in $O(1)$ from an analytical formula."

Please add the formula for the percentile.

**[L366]** "Riemann sum with $\sim 5000$ intervals."

Please justify accuracy (e.g., relative error $<1\%$ across all runs) and note the integrand's regularity. Did you try an adaptive quadrature (e.g., Simpson/Gauss–Legendre)?

**[L390]** "This formulation is useful in settings where quality requirements are clear but utilities are hard to quantify, for instance, when "good enough" responses are well-defined but the value of marginal improvements is ambiguous."

I find this hard to parse. Could you give an example?

**[L401]** "we use … FsfairX-LLaMA3-RM-v0.1 and RM-Mistral-7B."

Why these two reward models? Is there any reason for that?

**[L405]** "We always fix the max utility $B=1$."

Add one line that explains it. Something like: "by linearity, scaling utilities by $s>0$ and costs by $s$ leaves the threshold unchanged (since $\mathbb{E}[(U-\tau)_+]=c$ scales on both sides), so $B$ can be set to 1 w.l.o.g."

**Secretary problem mention**

Briefly mention the secretary problem as a classical optimal-stopping foil (unknown distributions, no recall), to position Pandora/Weitzman relative to other classics.

**Add general optimal-stopping references**

Like "Chow, Robbins, Siegmund (1971) *Great Expectations: The Theory of Optimal Stopping*" and maybe a modern set of lecture notes such as https://www.math.ucla.edu/~tom/Stopping/Contents.html. These will help readers new to the area.

**Fair-cap intuition**

Consider adding a one-line intuition to the fair-cap definition, something like: "$\tau$ is the reward level at which I'm indifferent between stopping now and paying cost $c$ for one more draw; i.e., the expected improvement from one more sample equals $c$."

---

> ### Author Response · Authors · 2025-11-22
>
> Thanks for the review! We've attempted to address the most salient comments below. Please let us know if further questions arise.
>
> **Corollary 7 and "vanishing regret" need reconciliation.** We thank the reviewer for pointing this out. As we haven't defined regret, we will remove this phrasing from the final version. That said, typically, when one states "vanishing regret" this is referring to the asymptotic behavior of the *normalized* regret (i.e. after dividing by the time horizon). Indeed, since our unnormalized sub-optimality gap is a constant, dividing by the time horizon will produce a vanishing "regret bound."
>
> **Heavy reliance on modeling assumptions; robustness is unclear.** We thank the reviewer for this question. We would like to point out that i.i.d draws per prompt is not an assumption; in Best-of-$N$ we often query the base LLM repeatedly, fixing the prompt and sampling procedure. Moreover, in practice, the per-sample cost of doing a forward pass through the LLM and reward model can be computed. Regarding the exponential tail-fit, we have uploaded a revised manuscript with QQ-plots showcasing the goodness of fit of the exponential distribution to the exponential rewards in Appendix F.6. That said, as we point out, theoretically, one must make some assumption about $F$ in order to get meaningful guarantees. Furthermore, our general framework extends to other distribution families beyond the exponential; we view the choice of distribution family as one that is up to the user, based on domain knowledge as well as iterative experimentation. We chose the exponential family as our illustrative example merely because it is particularly simple to analyze, and happens to do well on the datasets we explored. We like the reviewer's suggestion regarding quantifying the degradation under a controlled misfit, but due to the time constraint of the rebuttal period, we are unable to perform these experiments. Note that in Appendix F.5, we evaluate the performance of our method for different gold-standard quantiles $\alpha$, and found consistently good performance. This highlights the robustness of our method to at least the choice of $\alpha.$ Moreover, the QQ plots in Appendix F.6 show that while the Exponential distribution generally well-fits the right-tail of the exponentiated rewards, it is not perfect, and sometimes underestimates the true right tail. Despite this, we observe strong performance for these prompts, indicating some form of robustness to misspecification.
>
> **Discussion/conclusion are thin.** We thank the reviewer for this comment and will make sure to expand on the discussion and conclusion in the final version.
>
> **Important recent work on inference-time alignment and adaptive compute is missing.** We thank the reviewer for pointing out these references. We have actually cited a few of these in the Introduction and Related Works sections. That said, we will make sure to do a thorough comparison of them in the final version. For a detailed comparison, please see our response to Reviewer FJHZ.
>
>
> We will make sure to address all presentation issues in the final version.

---

### Official Review · Reviewer_FJHZ · 2025-11-03

**Soundness:** 2
**Presentation:** 3
**Contribution:** 2
**Rating:** 2
**Confidence:** 4

**Summary:**

* Problem: Large language model (LLM) generation requires balancing output quality against inference-time cost. Current strategies like Best-of-N sampling fix the number of generations in advance, leading to inefficiency.
* Solution: The paper reframes LLM inference as a Pandora’s Box optimal stopping problem. It introduces (1) a UCB-style Pandora’s Box algorithm that learns when to stop sampling without knowing the underlying reward distribution, and (2) a Bradley–Terry–based reward normalization to address cross-prompt scaling.
* Evaluation: Experiments on AlpacaFarm and HH-RLHF datasets across four LLMs and two reward models demonstrate that the adaptive method achieves comparable reward to Best-of-N sampling while using 15–35% fewer generations, supported by theoretical regret bounds relative to Weitzman’s optimal policy.

**Strengths:**

1. Novel framing: Establishes a principled connection between inference-time optimization and the Pandora’s Box optimal stopping problem which is an original theoretical perspective.
2. Theoretical grounding: Provides regret bounds relative to Weitzman’s optimal stopping policy and extends these results to exponential families.
3. Empirical consistency: Results are coherent across models, showing that adaptive inference can yield similar reward quality with reduced sampling.

**Weaknesses:**

1. A single basic baseline: The paper only compares against basic non-adaptive Best-of-N, omitting adaptive or GenRM/self-evaluation Best-of-N methods (e.g., speculative rejection, self-consistency, GenRM, self-evaluation, and many other approaches) that have surfaced in the past year or two. This limits the empirical significance of the reported 15–35% improvement which is relatively modest compared to what other works claim.
2. Lack of real compute and latency measurement: Efficiency is measured purely in terms of sample count, ignoring the additional cost of reward model inference or the sequential sampling latency. Parallel Best-of-N can often amortize cost, whereas the proposed adaptive approach incurs sequential latency. FLOPs or wall-clock time should be measured for fairness.

**Questions:**

1. Could the authors compare against recent adaptive test-time compute allocation methods (e.g., Wang et al. 2025, Sun et al. 2024, Manvi et al. 2024, etc.)?
2. Does the adaptive stopping decision depend on sequential generation, or could rewards be computed in parallel after sampling a batch?
3. How sensitive is the algorithm to the chosen percentile α (e.g., 0.99)? Would smaller values degrade performance significantly?

---

> ### Author Response · Authors · 2025-11-21
>
> Thanks for the review! We've attempted to address the most salient comments below. Please let us know if further questions arise.
>
> > Lack of real compute and latency measurement: Efficiency is measured purely in terms of sample count, ignoring the additional cost of reward model inference or the sequential sampling latency.
>
> We refer the reviewer to our general comment above regarding the batched variant of our method. In summary, our framework can be easily generalized to exploit parallel generation by defining a ``sample" from the reward distribution to be the maximum reward over a batch of rewards.
>
> > Could the authors compare against recent adaptive test-time compute allocation methods (e.g., Wang et al. 2025, Sun et al. 2024, Manvi et al. 2024, etc.)?
>
> Unfortunately, existing adaptive test-time compute allocation methods are not directly comparable. For example, adaptive self-consistency and self-evaluation do not involve an external reward model that serves as the proxy for answer quality, whereas our method does. Speculative rejection by Sun et. al. does not incorporate the actual cost of performing a forward pass through both the LLM and RM, and also requires the ability to compute rewards on partial sequences in order to preemptively prune away undesirable generations. In contrast, we explicitly incorporate the cost of a forward pass and only require the ability to evaluate the reward of the final response. Finally, the approach by Manvi et al. requires an external calibration dataset to train an auxiliary model for predicting the probability that the model cannot generate a better response than what it has so far.  In contrast, our method does not require an auxiliary dataset or model, and relies solely on the base LLM and RM itself. Finally, the reviewer references GenRM [1] as a type of adaptive Best-of-N method. However, this is incorrect as GenRM is actually just a reward model that can be used with BoN. Overall, we had trouble finding existing work that: (1) explicitly considered an inference cost c and (2) relied only on the base LLM and RM for determining when to stop sampling. That said, in Appendix F.7 of our revised manuscript, we have included a simple baseline which tracks the max reward so far and stops if it has not changed in some time. We find that this method performs consistently worse than our adaptive method.
>
> > Does the adaptive stopping decision depend on sequential generation, or could rewards be computed in parallel after sampling a batch?
>
> We refer the reviewer to our general comment above regarding the batched variant of our method.
>
> > How sensitive is the algorithm to the chosen percentile (e.g., 0.99)? Would smaller values degrade performance significantly?
>
> This is a great question. We have uploaded a revised version of our manuscript with experimental results for $\alpha = 0.50$ and $\alpha = 0.75$. These can be found in Appendix F.5 of the revised manuscript. We observe strong performance even for these value of $\alpha$, highlighting the robustness of our method to the reward transformation.

---

> > ### Comment · Reviewer_FJHZ · 2025-11-28
> >
> > Thank you for the detailed responses, the additional experiments on batching, and for the interesting and principled framework you have introduced. However, I will maintain my score as I remain unconvinced regarding the practicality of the method for the target edge deployment settings. Even with the proposed batched variant, the process remains fundamentally sequential (batch-by-batch) rather than fully parallel, which introduces latency bottlenecks that a fully parallel Best-of-N avoids. Furthermore, the high sample counts used in experiments (N > 50) appear unrealistic for resource-constrained edge deployment, and the continued lack of wall-clock time measurements makes it difficult to verify any tangible efficiency gains in a real-world deployment compared to fully parallel methods. Finally, I find the justification for omitting adaptive consistency baselines unconvincing; the fact that such methods function without the computational overhead of an external reward model makes them a highly relevant and arguably preferable alternative for efficient inference, as they eliminate an entire source of latency and cost.

---

> > > ### Author Response · Authors · 2025-11-28
> > >
> > > Thanks for the response! We agree that processing batch by batch incurs additional latency, and thus there is a trade-off between latency and adaptivity. Note that this trade-off is inherent to the problem: in order to be adaptive, one must pay an additional latency cost. Our batched variant allows one to interpolate along this trade-off via the choice of batch size. If the batch size is chosen to maximize the device’s memory, this is equivalent to a non-adaptive parallel BoN. On the other hand, if the batch size is set to 1, this corresponds to the fully adaptive approach described in the main text.
> > > Regarding our omission of adaptive consistency baselines, we believe these methods are not applicable to the tasks we consider for a couple reasons. First, unlike domains such as math, for general question-answering or alignment datasets it is unclear what “consistency” means or how it should be measured. Second, our objective is defined with respect to maximizing a known reward model. If this reward model is not used during inference, there is no guarantee that the most “consistent” output maximizes the reward.

---

### Author Response · Authors · 2025-11-21
**Addressing some general concerns**

We thank all reviewers for their thoughtful comments! Below, we provide a higher-level perspective of our contributions and address some general concerns.

**The Pandora's Box Framework as a Modular Design**: Reflecting on the reviews, we wish to emphasize that our contribution is not limited to a single algorithm but rather establishes a flexible *optimal stopping framework* for inference-time compute. This framework allows practitioners to optimize three key design axes:
  - *Distributional Family:* While our current implementation leverages the Exponential distribution (supported by the aforementioned QQ-plots), the underlying logic adapts to any distribution family where tail statistics can be estimated. In the next version, we plan to include ablation studies for alternative heavy-tailed distributions, such as the Lognormal distribution.

 - *Reward-Cost Alignment:* The critical prerequisite for optimal stopping is mapping abstract reward scores and compute costs onto a unified scale. This mapping is modular rather than fixed. To demonstrate this flexibility, Appendix F.5 of our revised manuscript includes results using Bradley-Terry transformations with varying gold-standard quantiles ($\alpha \in \{0.5, 0.75, 0.99\}$). We are also finalizing a comparative ablation study using a CDF-based transformation to further validate the framework's adaptability.

 - *Computational Granularity:*  As noted in our response regarding parallelization, the "atomic unit'' of generation is a flexible parameter. Practitioners can define a box opening at the granularity of a single sequence or a larger batch, depending on the memory architecture.

  In the final version, we plan to revise the paper to explicitly articulate these design flexibilities, highlighting the framework's adaptability to diverse deployment constraints.

**General Concerns**
- *Sequential vs. Parallel Inference:* Several reviewers were concerned about the fact that our method, as stated, is purely sequential and not parallelized (unlike vanilla BoN). We point out that one can easily adapt our method to handle batched/parallel queries -- instead of defining a sample as the reward for a single generated response, define a sample as the maximum reward over a batch of generated responses. We have uploaded a revised manuscript with experimental results for this batched version. As a final comment, we would like to emphasize that our adaptive stopping framework can interpolate between pure sequential and parallel generation based on the maximum batch-size your device can handle. From this perspective, our current experiments lie in the regime where only a single response can fit in memory, which is relevant for on-device/edge computing.
- *Empirical Validation of Tail Assumptions:* Several reviewers have asked for experimental verification for our choice to model the right-tail of the exponentiated rewards using an exponential distribution. We have uploaded a revised manuscript that includes some QQ-plots for prompts from the AlpacaFarm dataset to highlight the goodness of fit. These plots are included in Appendix F.6 and show that the exponential distribution is indeed a good fit for the tail behavior of the exponentiated rewards. Similar results are observed for other prompts in AlpacaFarm and HH-RLHF datasets as well as the FsfairX-LLaMA3 reward model. We will make sure to include these in the final version.

---

### Author Response · Authors · 2025-11-21
**Revised Manuscript Uploaded**

We have uploaded a revised manuscript. Changes (all of which are in the Appendix) are marked in blue.

---

### Meta-Review · Area_Chair_EdiV · 2026-01-05

**Summary:**

The submission proposes a novel framing of Large Language Model (LLM) inference as a Pandora’s Box optimal stopping problem, introducing a UCB-style algorithm to decide when to stop sampling based on observed rewards and estimated costs. While the reviewers praised the principled theoretical connection and the originality of the framework, the consensus for rejection is primarily driven by concerns regarding practicality and deployment viability. Specifically, reviewers pointed out that the method’s fundamentally sequential nature introduces significant latency bottlenecks compared to standard parallel Best-of-N sampling. Furthermore, the experiments relied on very high sample counts (often N > 50), which reviewers felt were unrealistic for standard production environments, and the omission of adaptive consistency baselines (which do not require external reward models) was seen as a significant gap in the empirical evaluation.

**Reviewer Concerns:**

During the rebuttal, the authors were highly proactive, providing a batched variant of their algorithm to address latency concerns and adding QQ-plots to empirically justify their exponential tail assumptions. They also included a stagnant reward baseline to bolster the empirical section. However, several critical concerns remain outstanding. Reviewers remained unconvinced that the "batched-sequential" approach offers a tangible wall-clock time advantage over fully parallel methods, noting that any degree of adaptivity in this framework still requires multiple sequential steps. Additionally, the lack of a head-to-head comparison with non-reward-model-based adaptive compute methods (e.g., adaptive self-consistency) remains a point of contention, as these alternatives might offer better efficiency by eliminating reward model overhead entirely.

**Reviewer Scores:**

Reviewer N324 is the most positive reviewer, they appreciated the theoretical novelty. Had the discussion continued, they likely would have maintained their score, as their concerns regarding reward normalization were largely addressed. Reviewer FJHZ remained the strongest advocate for rejection. Given their firm stance on the inherent latency-adaptivity trade-off and the perceived lack of practical edge-case utility, it is unlikely their score would have moved above a 3. Reviewer pf2g acknowledges the authors' efforts, but this reviewer maintained that the method's complexity and high inference cost (N \approx 300) limit its general usefulness. They might have moved to a 5 if more "small-model" or "real-world latency" experiments were provided, but remained a "marginal reject". Reviewer 7q3p was controversial due to potential AI generation and is focused on technical presentation. If the AI concerns were fully set aside, their score likely would have stayed at 4 or 5, as they felt the discussion/conclusion sections remained weak despite the rebuttal.

---

### Decision · Program_Chairs · 2026-01-26

Reject